# Molecular-level architecture of *Chlamydomonas reinhardtii's* glycoprotein-rich cell wall

Alexandre Poulhazan[1], Alexandre A. Arnold[1], Frederic Mentink-Vigier [2], Artur Muszyński [3], Parastoo Azadi[3], Adnan Halim[4], Sergey Y. Vakhrushev [4], Hiren Jitendra Joshi [4], Tuo Wang [5] ✉, Dror E. Warschawski [6] ✉ & Isabelle Marcotte [1] ✉

Microalgae are a renewable and promising biomass for large-scale biofuel, food and nutrient production. However, their efficient exploitation depends on our knowledge of the cell wall composition and organization as it can limit access to high-value molecules. Here we provide an atomic-level model of the non-crystalline and water-insoluble glycoprotein-rich cell wall of *Chlamydomonas reinhardtii*. Using in situ solid-state and sensitivity-enhanced nuclear magnetic resonance, we reveal unprecedented details on the protein and carbohydrate composition and their nanoscale heterogeneity, as well as the presence of spatially segregated protein- and glycan-rich regions with different dynamics and hydration levels. We show that mannose-rich lower-molecular-weight proteins likely contribute to the cell wall cohesion by binding to high-molecular weight protein components, and that water provides plasticity to the cell-wall architecture. The structural insight exemplifies strategies used by nature to form cell walls devoid of cellulose or other glycan polymers.

Plants and algae are efficient producers of biofuel precursors, textile and paper fibers, pharmaceutical compounds, and food supplements[1–3]. A central source of these products is the carbohydrate-rich cell wall, commonly found in both land plants and microalgae, by which it regulates cell integrity and many other physiological processes such as stress response[4]. Microalgae are a low-cost green production platform with rapid growth and facile scalability, ease of genetic modification, able to grow under hetero- or phototrophic conditions. As eukaryotes, they also offer a wide post-translational modification machinery used for recombinant protein expression and synthetic biology[5,6]. However, to better exploit microalgae for industrial applications, we imperatively need

an in-depth understanding of the molecular-level architecture of their cell wall[7].

*Chlamydomonas reinhardtii* is a freshwater microalga and stands out as one of the most abundantly studied. It is a model organism for directed mutagenesis applied to different research fields including biochemistry, physiology, and genetics[8,9]. In 2007, the fourth complete report of the *Chlamydomonas* genome revealed that it shares 706 protein families with humans[10]. The cell wall of *C. reinhardtii* has been investigated mainly from the 1970s to the late 2000s[11,12]. In contrast with the cellulose-based cell walls widespread amongst microalgae, that of *C. reinhardtii* is built on glycoprotein-rich layers reminiscent of extensins in higher plants[13]. Specifically, it is composed

[1]Department of Chemistry, Université du Québec à Montréal, Montreal, QC H2X 2J6, Canada. [2]National High Magnetic Field Laboratory, Florida State University, Tallahassee, FL 32310, USA. [3]Complex Carbohydrate Research Center, University of Georgia, Athens, GA 30602, USA. [4]Copenhagen Center for Glycomics, University of Copenhagen, Copenhagen, Denmark. [5]Department of Chemistry, Michigan State University, East Lansing, MI 48824, USA. [6]Laboratoire des Biomolécules, LBM, CNRS UMR 7203, Sorbonne Université, École Normale Supérieure, PSL University, 75005 Paris, France. ✉e-mail: wangtuo1@msu.edu; dror.warschawski@sorbonne-universite.fr; marcotte.isabelle@uqam.ca

of five dense glycoprotein-rich layers that form a protective shell, as seen by electron microscopy (EM)[14–17]. The water-insoluble core of the cell wall is made of three central layers, named W4/W6/W4, that contain fibrous material and denser objects described as granules[18,19] (Fig. 1a). According to previous studies, this central triplet comprises a large number of proteins, including high molecular weight hydroxyproline-rich glycoproteins (HRGPs), i.e., GP1 (~300 kDa), GP1.5 (170 kDa), GP2 (155 kDa, 110 kDa after deglycosylation) and GP3 (135 kDa, 65 kDa once deglycosylated)[16,20]. Low molecular weight proteins ranging from 30 kDa to 50 kDa referred to as "14-3-3" are also found and believed to act as cell wall cross-linking precursors[19–22]. These "14-3-3" proteins are rich in serine (Ser) and threonine (Thr), but not fully sequenced[22]. The remainder of the cell wall, also called the "insoluble cell wall" since it cannot be extracted even using chaotropic agents, is composed of a membrane-anchoring protein layer (W1), a dense layer with fibrillar proteins (W2), and an outer fibrillar surface (W7)[14,20]. The molecular mechanisms by which all these various glycoproteins interact are still unknown.

Previous investigations of *C. reinhardtii* cell wall provided a partial sequencing and composition of HRGPs by chromatography and mass spectrometry (MS), while EM revealed the layered architecture. HRGPs were shown to be glycosylated by short O-oligosaccharides containing 1–5 residues in which arabinose (Ara) and galactose (Gal) are the most abundant. These oligosaccharides would be linked to HRGP domains containing Ser or polyproline (PP) helices[19,23–25]. Traces of mannose (Man) were also reported, accounting for up to 7% of the overall glycan content, as well as β(1-2)-linked L-Arabinose (Ara) disaccharides substituted with galactofuranoses (Gal*f*) and O-methylation[26]. While O-glycosylation is the only post-translational modification described so far in the cell wall, an N-glycosylation pathway has also been described in *C. reinhardtii* that involved the linking of two N-acetylglucosamines (GlcNAc) and two Man to an asparagine (Asn)[27].

Here we use solid-state nuclear magnetic resonance (ssNMR) to determine the chemical nature and dynamics of the different water-insoluble but chaotrope-soluble cell wall constituents at the molecular level, without compromising its integrity since no extensive purification or digestion of the extracted cell wall is required. We establish the complete amino acid and glycan profile and reveal preferential contacts between carbohydrates and proteins. We also expose the presence of conformational domains and potential O- and N-glycosylation on the proteins, as well as the abundance of Man residues corresponding to low molecular-weight glycoproteins (LWGPs). We show that oligosaccharides are short, and reveal the segregation of glycan- and protein-rich domains with different hydration and dynamics. This work provides a molecular-level refinement of the complex protein-rich cell wall architecture of an aquatic microorganism in which glycans play a key role, and highlights the role of water in this superstructure.

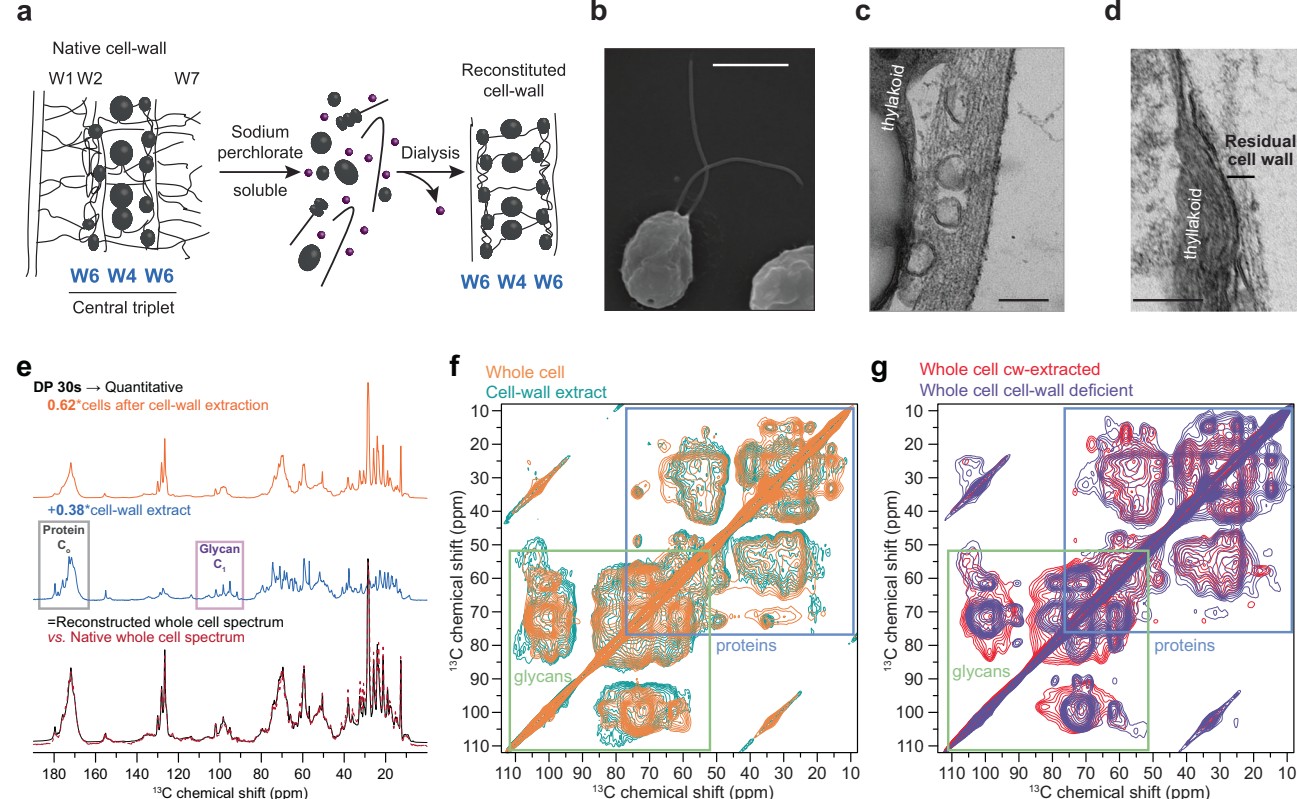

**Fig. 1 | Cell wall extraction of *C. reinhardtii* preserves carbohydrates and cell surface. a** Previously described layered organization and extraction protocol to reconstitute the native central triplet of the cell wall, based on the overall assembly observed by EM[19] (adapted from Goodenough et al.[19], used with permission of Rockefeller University Press; permission conveyed through Copyright Clearance Center, Inc). **b** SEM image of whole *wt C. reinhardtii* cells (scale bar: 5 µm). **c** TEM image of *C. reinhardtii* cell surface before cell wall extraction reveals a layered structure and large spherical components or cavities (scale bar: 200 nm). **d** TEM image after wall extraction shows remnants on the cell surface (scale bar: 200 nm). **e** 1D [13]C ssNMR of the cells after extraction (top; orange), cell wall extract (middle; blue), and intact cells (bottom; red dash line). The weighted sum (bottom; black) of the cell wall extract and the cell residual matches the spectrum of the intact cells, indicating no loss during the extraction protocol and that the cell wall accounts, here, for 38% of the overall intensity (matching sample reconstitution performed using Table 1). **f** Similarity of the intact whole-cell MAS-DNP (orange) and extracted cell wall (teal) spectra. **g** MAS-DNP spectra of *wt* cells after wall extraction (red) and cell-wall deficient *cw15* strain (purple) show that the central triplet, absent on the *cw15* strain, is extracted in the *wt* strain. MAS-DNP [13]C-[13]C DARR experiments were recorded on a 600 MHz/395 GHz spectrometer at 100 K and 8 kHz MAS frequency. SEM/TEM (panels **b**–**d**) images were recorded in triplicate on independent biological replicates giving similar results.

## Results

### Cell wall extraction retains the macromolecular core

We extracted *C. reinhardtii*'s water-insoluble (but chaotrope-soluble) cell wall constituents and verified if their composition was identical to intact cell walls. The cell wall was extracted with chaotropic agents, then slowly reconstituted after dialysis. We employed a milder protocol with fewer dialysis steps compared to the original method[18,20], to preserve the minor constituents of the cell wall (Fig. 1a). Scanning electron microscopy (SEM) (Fig. 1b) and transmission electron microscopy (TEM) (Fig. 1c) respectively show the intact microalga and the characteristic layered organization of the cell wall described in the literature. Cells were also undamaged after wall extraction, as demonstrated by the intact membrane bilayer and internal organelles such as thylakoids, partially visible in Fig. 1d. This was further confirmed by flow cytometry analysis, which revealed that about 80% of the cells remained alive and metabolically active after wall extraction.

The composition of the cell wall extract was also compared to intact cells using both the standard 1D ssNMR (Fig. 1e) and Magic-Angle Spinning Dynamic Nuclear Polarization (MAS-DNP) (Fig. 1f, g), which uses microwaves and a polarizing radical to provide a tremendous increase in sensitivity[28,29]. Here, we observed a 45-fold enhancement in NMR signals (Supplementary Fig. 1) but a loss in spectral resolution due to the low temperature required by this method. The weighted sum of the two spectra recorded on the cell wall extracts and the remainders after extraction correspond exactly to the spectrum of the intact cell walls (Fig. 1e), indicating no loss during our extraction protocol.

The 2D $^{13}$C-$^{13}$C correlation spectra of both intact *C. reinhardtii* cells and extracts of cell walls exhibit a high degree of similarity. This is evident in the overlap of MAS-DNP spectra at 100 K, which show signals exclusively from the outer region of the cell (Fig. 1f), as well as in the data collected at higher temperature (283 K), which encompass all cellular components (Supplementary Fig. 2). This congruence confirms that the preservation of cell wall structure in the extract. It also suggests that the polarizing radical does not diffuse through the membrane but preferentially localizes in the cell wall. The minor differences can be ascribed to the fact that only the W4 and W6 layers were removed when extracting the cell wall, leaving the fibrillar W1, W2 and W7 layers attached to the cell surface (Fig. 1a). Because MAS-DNP only enhances and detects signals from the cell surface constituents, the intensity is thus improved, explaining for instance, the glycan-protein intermolecular contacts seen on Fig. 1f. This was also confirmed by the similarities between the MAS-DNP spectra of the cell-wall deficient strain (*cw15*) and the wild-type (*wt*) strain after cell-wall extraction (Fig. 1g). The *cw15* mutant has compromised cell wall synthesis[8]. The remaining wall components resemble the cell-wall outer layer[30], resist chaotropic extraction[31] and therefore likely correspond to the residual chaotrope-insoluble W1, W2 and W7 layers. The resemblance between the MAS-DNP spectra of *cw15* cells and the *wt* sample after cell-wall extraction further confirms that our extracts are composed of the cell wall central triplet.

### Glycan composition of the cell-wall extract

We determined the complete glycan and amino acid profile in *C. reinhardtii*'s cell wall extracts. We used the flagella-deficient *bald2* strain to avoid the occurrence of large flagella-associated proteins (~200 kDa) that could contaminate cell-wall extracts during the growth cycle. Our results were independent of the strain type, i.e., with (*wt*) or without (*bald2*) flagella, the growth conditions or cell differentiation (sexually competent, differentiated or not), as revealed by the protein composition of different cell wall extracts obtained using polyacrylamide gel electrophoresis (Fig. 2a) or ssNMR (Supplementary Fig. 3 and Supplementary Table 1). The amino acid profile of the protein bands separated by SDS-PAGE was also analyzed using HPLC on

each cut band. The unlabeled and $^{13}$C-labeled cell-wall protein profiles were also comparable using these methods.

Using fully relaxed 1D $^{13}$C ssNMR spectra, we compared the integrals of peaks characteristic of carbohydrates (anomeric $C_1$) or proteins (carbonyl CO) (Fig. 1e; middle row). A 1:3 glycan/protein molar ratio was determined, indicating that a considerable part of the cell wall is composed of glycans, in agreement with the literature[27]. This ratio can reach 1:2 because of the large proportion of Gln/Glu and Asn/Asp - revealed by the amino acid analysis described below - that comprise two carbonyl groups. The composition of the cell wall extracts was then monitored using SDS-PAGE and compared to the literature[14]. The amino acid analysis of the SDS-PAGE bands of the native cell wall confirms that the high molecular weight proteins in cell wall extracts are rich in hydroxyproline (Hyp) and correspond to the reported HRGPs (Table 1). We found a higher abundance of low molecular weight proteins following our milder extraction protocol (Fig. 2a).

Quantitative analysis of the saccharide content and bonds of *C. reinhardtii*'s reconstituted cell wall using our protocol[32] was achieved using MS (Fig. 2b) and 2D $^{13}$C-$^{13}$C ssNMR spectra (Fig. 2c and Supplementary Fig. 4). We found a majority of the residues being Man (52%), followed by Gal (15%), Ara (15%), GlcNAc (11%), non-modified glucose (Glc) (3%), rhamnose (Rha) (3%) and xylose (Xyl) (3%), as reported in Fig. 2d. A significant proportion of Glc is N-acetylated (GlcNAc), as previously but not quantitatively reported in *Chlamydomonas* cells[33]. One-third of anomeric carbons ascribed to Man and Gal have very low chemical shift values, and can therefore be assigned to non-reducing terminal groups of oligosaccharides (pink bars on Fig. 2d)[34,35]. Their abundance indicates the presence of short oligosaccharides, with approximately two glycan units followed by a terminal glycan. In addition, no Ara but almost all Gal are identified as terminal groups, confirming that most Ara oligosaccharides would be terminated by a Gal residue[26]. These results are supported by the high proportion of non-reducing terminal groups detected by semi-quantitative GC-MS (see t-X glycans on Fig. 2b).

The GC-MS analysis of glycan linkages demonstrates the high variety of glycosidic bonds in *C. reinhardtii*'s cell wall (Supplementary Table 2). It also helps evaluate the efficiency of our cell wall extraction protocol by comparing the glycan composition of whole cells (dominated by starch Glc), cells after wall extraction, and the extracted wall (Supplementary Table 7), showing that about 40% of the cell's glycans are contained in the cell wall, and that almost no sugar was lost or altered during the extraction and reconstitution (Supplementary Table 8 and Supplementary Fig. 2).

A striking finding is the high abundance of Man and GlcNAc in *C. reinhardtii*'s cell wall, which has been debated in the literature[27,36]. While Ara and Gal are generally presented as the most abundant glycans in HRGPs[11,26,37], earlier work showed that a water-soluble fraction of *Chlamydomonas* cell wall was rich in Man[38] and, more recently, that GP3 in the cell wall was heavily mannosylated[24]. The presence of glycoproteins rich in Hyp interacting with Man-rich lectin-like domains involved in the cell wall of *C. reinhardtii* has also been demonstrated in cell-differentiated zygotes[17]. Finally, recent work reported oligomannose glycans and 6-O-methylmannose in soluble and membrane-bound proteins of *C. reinhardtii*[36], as well as two putative mannosyltransferases, consistent with the presence of mannosylation[39]. The Man depletion after cell wall extraction and in the *bald2* mutant (Supplementary Tables 7 and 8), together with the MAS-DNP detection of Man signals in the intact cell walls, are strong indications that Man-rich glycoproteins are part of the cell wall, or at least strongly associated with it.

### Amino acid composition and LWGP proteins

We then assigned and quantified the protein content using 2D $^{13}$C refocused INADEQUATE spectra and HPLC methods (Fig. 3a,

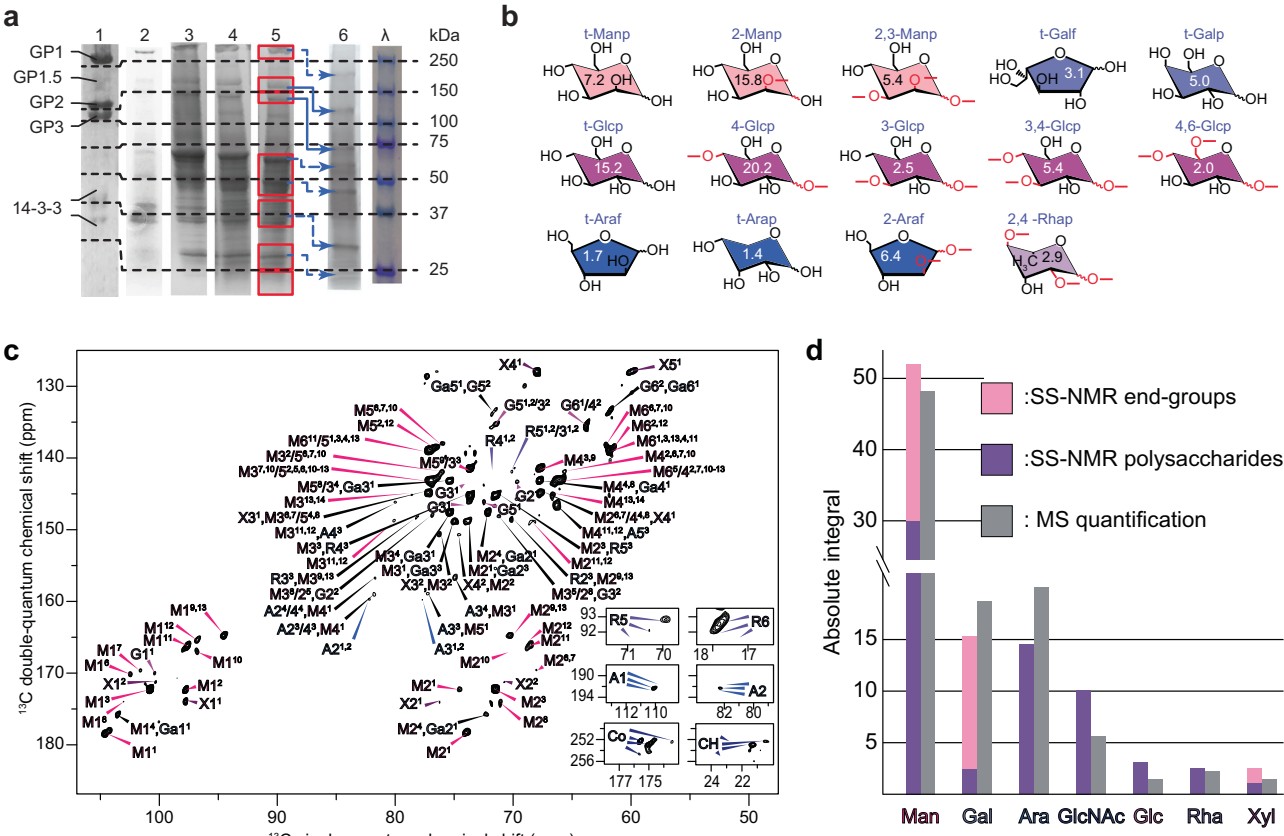

**Fig. 2 | *C. reinhardtii* cell wall composition analyzed by electrophoresis, MAS and ssNMR. a** SDS-PAGE (performed at least in triplicate and giving similar results) of cell wall extracts shows HRGPs and additional enrichment in low molecular weight proteins (<100 kDa). Lane 1: cell wall extract published by Goodenough et al.[14], used with permission; lane 2: cell wall extracted from the *bald2* strain; lane 3: cell wall extracted from sexually differentiated cells; lane 4: [13]C-labeled *wt* cell wall; lane 5: [13]C-labeled *bald2* cell wall used for ssNMR assays; lane 6: deglycosylated [13]C-labeled *bald2* cell wall with tentative molecular weight shift assignment; lane λ: molecular weight ladder. **b** Representative structures of glycosidic linkages identified by GC-MS (carbons involved in linkages shown in red). The numbers for each unit correspond to the percentage of such linkages obtained semi-

quantitatively (Supplementary Table 2). **c** [13]C-[13]C through-bond ssNMR INADE-QUATE spectrum of extracted C. *reinhardtii* cell walls used for carbohydrate and amino acid identification and quantification. A $XZ^Y$ glycan notation is used, where X is the type of sugar, Y the spin system number, and Z the carbon number (for example, Ga6[1] is the carbon 6 of the Gal unit number 1). Insets at the bottom right of the spectrum show the unique signals of Rha $C_6$ (top), Ara $C_1$ (middle), and acetyl/methyl group from GlcNAc (bottom). **d** SsNMR and GC-MS quantification of glycan units; End groups/oligosaccharides are in pink, and polysaccharides in purple, both differentiated by their $C_1$ chemical shifts (Supplementary Tables 3–6). Source data are provided as a Source Data file.

Supplementary Figs. 5–7 and Supplementary Tables 9–12). According to ssNMR, the most abundant amino acid residues in the cell wall extract are alanines (13%), followed by glutamines and glutamates (9% each), glycines (8%), asparagines and aspartates (6% each). We note that ~18% of the amino acids can be glycosylated. The relative abundance of the amino acids in the cell wall extract determined by HPLC, which is more sensitive and reproducible throughout different samples (Supplementary Table 12), is consistent with the ssNMR results for most amino acids (Table 1). No cysteine (Cys) or tryptophan (Trp) residues were detected by either technique, while histidine (His) or Hyp residues could not be detected by ssNMR, their abundance being below the 2% sensitivity threshold. Hyp was however detected using 2D [13]C-[13]C-PDSD and MAS-DNP at low temperature (see below and Supplementary Fig. 8) but could not be quantified.

Table 1 and Fig. 3c reveal that Hyp is abundant (up to 22%) in GP1 to GP3, consistent with previous studies reporting that this residue constitutes up to one third of the total amino acid content in HRGPs[12,14,22,37,40,41]. Its overall proportion, however, is very low (~3%) in whole *C. reinhardtii* cell walls since HRGPs constitute only ~20% of the total protein content, based on SDS-PAGE image treatment (Figs. 2a and 3b, Table 1). This is also true for other amino acids, such as Ser for example.

To eliminate all glycan units from the cell wall extract without degrading the proteins, we performed a chemical deglycosylation since it allows removing both O- and N-linked glycosylations[42]. The deglycosylation led to a shift of all SDS-PAGE protein bands, proving that the low molecular weight proteins are also glycosylated (Fig. 2a, lane 5 *vs.* 6), to an approximate level of 20 % assessed by the change in molecular weight.

We were thus able to identify four new groups of low molecular weight glycoproteins (LWGPs) that we will refer to as GP4 to GP7, which contribute to almost 80% of the amino acids content in our reconstituted cell wall, and have a very low Hyp content. The molecular weights of GP4 (~50 kDa) and GP5 (~37 kDa) correspond to those reported for the "14-3-3" protein fraction[21,22]. GP4 and GP5 contribute *ca.* 35% of the amino acids, while the two lighter protein bands, GP6 (~25 kDa) and GP7 (<20 kDa), account for the *ca.* 45% left. LWGPs as well as "14-3-3" proteins, could be considered as "non-canonical cell-wall proteins"[43,44], but they do survive the extraction protocol, two dialysis steps, renaturing non-chaotropic conditions and centrifugation. They should therefore be considered either as part of the cell wall, or strongly interacting with it. The identification of a greater variety of glycoproteins in our cell wall extract, without any significant cell damage, confirms that our protocol is milder, and

**Table 1 | Amino acid composition of the cell wall and specific proteins**

| | Total cell wall | | HRGPs | | | LWGPs | | | |
| | %mole | | GP1/1.5 | GP2 | GP3 | GP4 | GP5 | GP6 | GP7 |
| | | | 298/280 kDa | 165 kDa | 140 kDa | ~50 kDa | ~37 kDa | ~25-20 kDa | <20 kDa |
| | ssNMR | HPLC | %mole | %mole | %mole | %mole | %mole | %mole | %mole |
|---|---|---|---|---|---|---|---|---|---|
| Gln | 9.1 | 19.0 (0.3) | 7.2 | 5.6 | 19.5 | 17.3 | 18.7 | 23.0 | 15.3 |
| Glu | 8.8 | | | | | | | | |
| Asn* | 6.4 | 14.8 (0.3) | 5.6/4.7/9.9 | 7.0/8.1 | 2.5/1.6 | 10.2 | 14.3 | 19.5 | 12.9 |
| Asp | 5.9 | | | | | | | | |
| Ala | 13.1 | 10.9 (0.3) | 9.4/10.1/8.4 | 10.5/8.5 | 15.4/10.2 | 8 | 9.9 | 11.9 | 8.7 |
| Gly | 8.5 | 8.0 (0.3) | 11.6/6.6/23 | 7.8/8.5 | 10.7/11.0 | 8.2 | 10.3 | 6.5 | 10.8 |
| Leu | 8.0 | 5.5 (0.3) | 3.7/2.9/5.1 | 7.5/7.0 | 0.2/7.0 | 10.2 | 4.8 | 4.6 | 8.4 |
| Lys | 6.2 | 5.3 (0.1) | 2.7/3.6/4.2 | 3.0/3.0 | 7.1/4.5 | 5.8 | 6.5 | 3.4 | 8.2 |
| Ser* | 6.2 | 5.0 (0.2) | 14.4/15.8/10.7 | 7.0/8.3 | 8.1/9.3 | 3.3 | 5 | 3.2 | 5.3 |
| Pro | 5.8 | 4.8 (0.2) | 3.4/2.7/3.2 | 5.5/7.7 | 0.7/4.5 | 6.6 | 6.1 | 4.3 | 2.7 |
| Val | 5.7 | 5.6 (0.3) | 3.4/4.4/3.0 | 7.3/5.7 | 5.3/5.9 | 9.7 | 4.1 | 4.3 | 5.1 |
| Arg | 5.1 | 3.5 (0.3) | 1.9/1.7/2.7 | 3.4/3.0 | 3.6/2.1 | 3.8 | 4 | 2.4 | 5.5 |
| Thr* | 4.3 | 4.7 (0.4) | 5.3/4.2/4.3 | 6.7/6.6 | 7.7/7.4 | 2.5 | 3.9 | 3.8 | 4.3 |
| Met | 2.5 | 1.6 (0.2) | 1.1/0.8/0.5 | 1.6/1.0 | 1.4/0.9 | 1.6 | 2.7 | 1.3 | 1.9 |
| Ile | 2.2 | 3.1 (0.3) | 3.2/3.3/2.3 | 4.2/1.3 | 2.2/3.2 | 5.9 | 3.1 | 4.0 | 2.9 |
| Tyr | 1.5 | 2.4 (0.2) | 1.9/0.9/1.4 | 3.3/3.6 | 2.6/1.2 | 2 | 3.7 | 2.4 | 3 |
| Phe | 0.7 | 2.8 (0.4) | 2.2/1.3/2.3 | 4.4/3.8 | 6.4/3.6 | 3.1 | 1.5 | 2.8 | 3.5 |
| His | NA | 2.0 (0.1) | 0.6/0.8/1.5 | 0.8/0.2 | 1.0/1.0 | 1.4 | 1.3 | 1.5 | 1.4 |
| Hyp* | NA | 1.0 (0.1) | 22.4/32.3/15.5 | 14.4/14.7 | 6.6/5.7 | 0.4 | 0.1 | 1.1 | 0.1 |
| % SDS-page | | | 5.3 | 7.2 | 9.3 | 10.6 | 24.3 | 35.1 | 8.3 |
| | | | 21.8 | | | 78.2 | | | |

Amino acid composition of the cell wall determined by ssNMR and HPLC (values in brackets correspond to standard deviation of three different HPLC samples) (see Supplementary Table 12 for replicates), and of specific proteins separated by SDS-PAGE and analyzed by HPLC. Published values are indicated for comparison and underlined[19,24]. Relative protein proportions (bottom two lines) were evaluated on Coomassie blue stained SDS-PAGE. The asterisk (*) symbol indicates amino acids that can be glycosylated.

is also an indication that its protein composition is closer to that of the native microalga.

To summarize, our results show that the amino acid content in the cell wall extract is dominated by Gln/Glu, Asn/Asp, Ala and Gly, with the lower weight GP4 to GP7 enriched in Gln/Glu and Asn/Asp. Interestingly, with the exception of Gln/Glu, these abundant residues have been reported to play an important role in plant cell walls[45]. Asparagine is known to be involved in N-linked glycosylation, while Ala-rich and Hyp-containing cell walls, possibly associated with arabinogalactan, have been reported in plants and fungi[46]. Also, Gly-rich proteins are involved in the formation of β-pleated sheet structures important for cell mechanical resistance in plant cell walls[47]. Altogether, *C. reinhardtii* probably shares some structural features with plants in terms of amino acid composition of cell wall proteins.

### β-sheet domains, protein-glycan and inter-glycan contacts

The protein secondary structure was probed by analysis of the amino acids' $^{13}C$ chemical shifts, which are sensitive to the conformational environment. We could find no evidence of α-helices, but identified about half of the residues in β-sheet environments, mostly Ile, Lys, Thr and Val. The remaining chemical shifts were compatible with either random coils or PPII helices[20], principally for Asn/Asp, Gly, Pro and Ser (blue bars on Fig. 3a and Supplementary Table 9). No correlation could be established between the hydrophobicity or charge of the residues, or their secondary structure propensity. Our results do not clearly indicate but cannot rule out the presence of PPII that have been reported in HRGPs[23,48].

We then explored the role of glycans in the structuration of cell wall proteins by recording ssNMR spectra of deglycosylated cell walls. While the complete deglycosylation resulted in a total disappearance of the glycan signals in the 2D INADEQUATE (Fig. 4a) and DARR spectra (Supplementary Fig. 9), the amino acid chemical shift values remained almost identical, indicating no significant change in secondary structure of the proteins. However, Hyp and Thr signals were completely lost following deglycosylation, which could be explained by an increase in their mobility after deglycosylation, going from a "slow motion regime" to the ssNMR unfavorable "intermediate regime". This is a strong indication that, in addition to Hyp, Thr residues are heavily glycosylated to unidentified glycans in *C. reinhardtii* cell walls - another information that has never been reported before. It is possible, however, that Hyp- and Thr-rich regions could be structured before deglycosylation, but this change would not be detectable due to their signals' disappearance. Altogether, glycans appear to have almost no influence on the overall protein structure and dynamics, suggesting a model in which they are spatially isolated from the amino acids, in sugar domains outside or within protein segments. This information must be considered in building a new cell wall model.

We pursued the examination of glycan through-space connectivities using dipolar-based ssNMR methods. Multiple glycan-glycan contacts were revealed by the $^{13}C$-$^{13}C$ CP-PDSD spectrum of *C. reinhardtii*'s cell wall (Fig. 4b). For example, we detected inter-Man contacts between their $C_{2/3}$ (76 ppm) and $C_1$ of units 9, 13 (94.5 ppm), 2, 10 and 12 (97.2 ppm). We also identified inter-glycan contacts between Man $C_1$ (94.5 and 97.2 ppm) and Ara $C_1$ (109.8 ppm), $C_2$ (82.2 ppm) and $C_{3/4}$ (77.4 ppm). These units are probably not covalently linked[27], but spatially close.

The spin-diffusion driven experiments (PDSD and DARR) did not allow detecting specific glycan-protein contacts (Supplementary Fig. 11), probably because of fast relaxation or phase separation in dynamically distinct domains. The increased sensitivity provided by

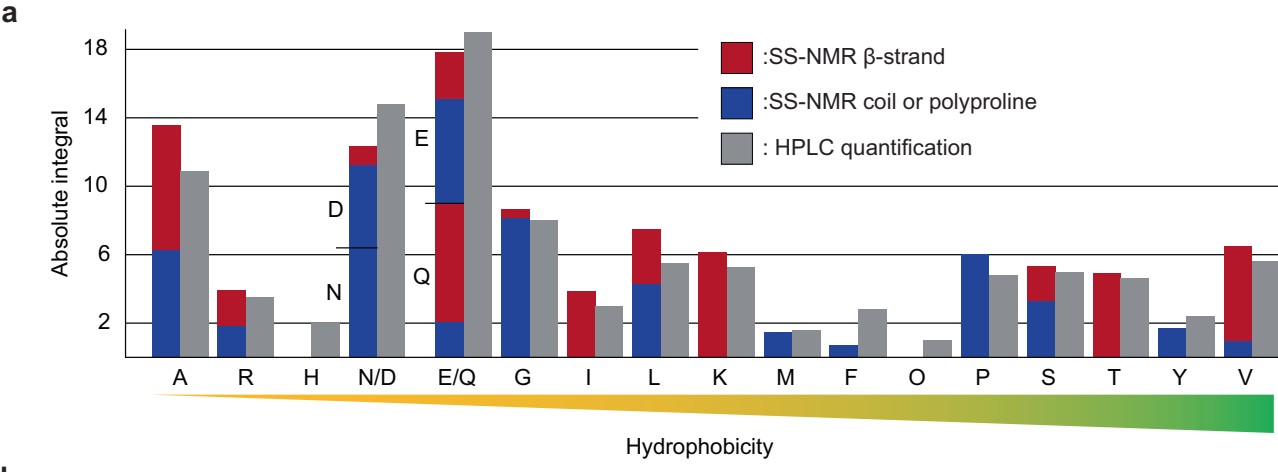

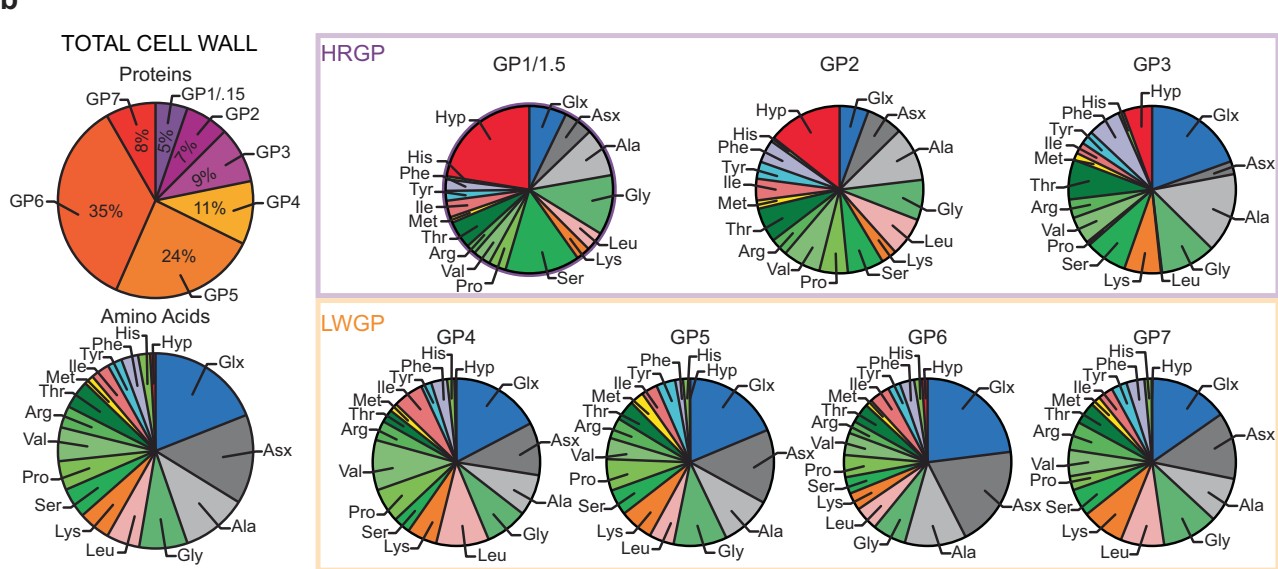

**Fig. 3 | *C. reinhardtii* cell wall protein composition analyzed by electrophoresis and ssNMR reveal low Hyp relative abundance. a** ssNMR (red/blue bars), and HPLC (gray bars, error <0.5% for all amino acids) quantification of amino acid units and ssNMR secondary structure prediction by TALOS+ (β-strands are in red, and random coils are in blue, full spectrum and numerical values are reported in Supplementary information). **b** Glycoprotein (GP) proportion and amino acid quantification in the cell wall extracts obtained from SDS-PAGE and HPLC, respectively. Source data are provided as a Source Data file.

hyperpolarization and low temperature enabled the detection of important glycan-protein contacts by MAS-DNP (Fig. 4c). As $C_1$ carbons have a relatively large chemical shift value and dispersion, they were used to identify glycans in the MAS-DNP spectrum. Since the spectral resolution is reduced by line broadening at cryogenic temperature, we used the assignment obtained at 283 K, assuming no major chemical shift differences[49,50]. These experiments, based on inter-spin dipolar coupling, only provides information on the spatial distribution, through-space contacts, but does not distinguish glycosylation via covalent bonds. We found contacts between Thr ($C\gamma$ at ~21 ppm) and terminal Man units 10 or 12 ($C_1$ ~ 95.3 ppm). We also observed contacts between all Hyp carbons (sidechain at ~35, 50, 59 and 73 ppm) and both Ara ($C_1$ ~ 110 ppm) and the terminal Man units 9 or 13 ($C_1$ at ~94.8 ppm). These connectivities might result from the specific geometry of the Man-Hyp contact or reflect a tighter binding with reduced mobility. Although the glycosylation of Ser is expected[13], Ser-glycan contacts could not be resolved due to the overlap of Ser $C\alpha$ resonances with those of Hyp and Thr, and that of Ser $C\beta$ resonances with glycans' C6. Other intense glycan $C_1$-amino-acid contacts could be detected for Glc or Rha ($C_1$ at 103.5 ppm), or Man/Gal ($C_1$ at 105.8 ppm), but the limited spectral resolution did not allow to identify the amino acid residues involved. Conversely, unidentified glycan $C_1$ atoms showed an intense contact with Asn's carbonyl by MAS-DNP PDSD with a long (1.5 s) mixing time (Fig. 4c, bottom).

The sensitivity enhancement provided by MAS-DNP also allowed us to record a $^{15}N$-$^{13}C$ ssNMR spectrum (Fig. 4d and Supplementary Fig. 13), and identify backbone nitrogen signals of amino acids, such as Hyp and Thr (-118 ppm) and Asn (-112 ppm). The one-bond NCα spectrum revealed a likely different unidentified glycan $C_1$ signal with an intense contact with Asn's amide (Fig. 4d) - a contact that was not detected at 283 K (Fig. 4d, purple spectrum). Using an additional $^{13}C$-$^{13}C$ mixing time in an NCaCx experiment, we allowed magnetization to transfer from Cα to surrounding carbons. Connections were thus revealed between O-glycosylated Hyp/Thr's amides (115 and 120 ppm) and several glycans' $C_1$ sites, including that of Ara (109 ppm) (Supplementary Fig. 13).

In summary, we detected unambiguous contacts between Hyp and Ara, Hyp and Man, Thr and Man, as well as Asn with unidentified glycan $C_1$s. While we could not quantify Ser or Asn glycosylations, the glycan-Hyp/Thr contacts and their relative intensities mostly identified using MAS-DNP DARR spectra, are summarized in Supplementary Fig. 10 and Supplementary Table 13.

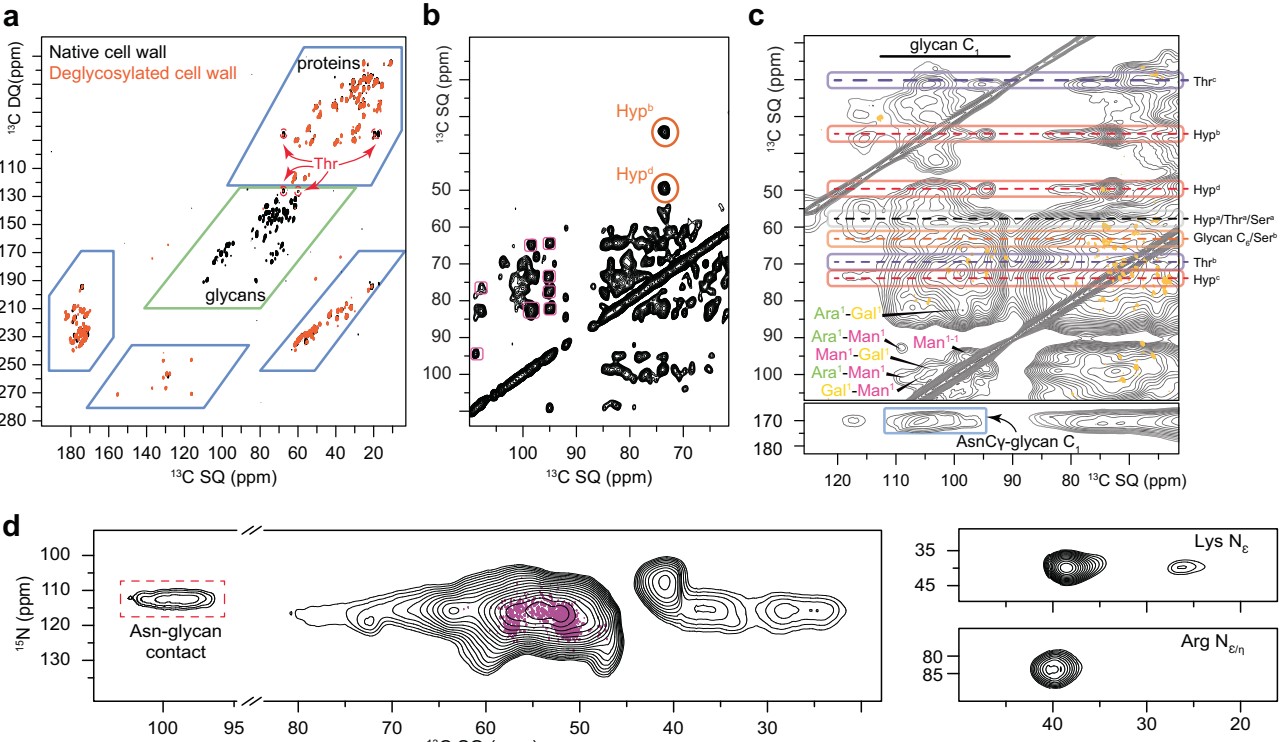

**Fig. 4 | Glycan-glycan and glycan-protein contacts revealed by ssNMR. a** The effect of deglycosylation on *C. reinhardtii* extracted cell-wall is probed by 2D through-bond $^{13}$C experiments, by comparing native (black) and deglycosylated (orange) cell wall extracts. **b** Hyp detection, not possible via through bond experiment, is enabled using a 100 ms-long through-space PDSD, and inter-glycan contacts are highlighted in pink. **c** MAS-DNP experiments, which probe spatial proximities (DARR 50 ms) (gray), allow glycan-protein contact detection using the assignment obtained at 298 K (orange). Contacts between proteins and glycan $C_1$ are circled and color-coded (Ara (green), Gal (yellow), and Man (pink)). **d** Asn

sidechain nitrogen (~113 ppm) covalently linked to GlcNAc $C_1$ (~100 ppm) involved in N-glycosylation are detected using a MAS-DNP enhanced N-C$\alpha$ correlation (black spectrum). These contacts are not detected at 283 K (purple spectrum recorded at 800 MHz). Additional amino acids with N in their sidechains are also be detected (two top panels). Unique glycan-to-Thr/Hyp contacts are listed in Supplementary Fig. 10. **a, b** NMR spectra are recorded at a field of 600 MHz and 298 K under 13.5 kHz MAS while **c, d** data come from MAS-DNP experiments at 600 MHz/ 395 GHz and 100 K under 8 kHz MAS.

The high abundance of Man in *C. reinhardtii*'s cell wall suggests the presence of O-glycosylation to Thr. To verify this hypothesis, a cell-wall protein extract was digested with trypsin and enriched by Concanavalin A lectin weak affinity chromatography (ConA-LWAC)[51,52]. Our glycoproteomic analysis identified 9 unique glycopeptide sequences from 6 proteins (Supplementary Table 14) using higher-energy collisional activation (HCD) and electron-transfer dissociation (ETciD) MS. Although none of the identified proteins is annotated as a cell wall protein, it should be noted that the approach used favors abundant, small, soluble proteins with simple glycosylation patterns. Nevertheless, these glycoproteomic results suggest an O-linked glycosylation of Ser and Thr with hexose modifications, likely α-linked Man or Man2, thus demonstrating the presence of biosynthetic machineries in Chlamydomonas capable of catalyzing these modifications.

Large cell wall proteins with complex glycosylation patterns are notoriously difficult to detect by the approach described above, without using specific molecular separation techniques, enrichment and targeted detection and analysis[53,54]. They are however amenable to ssNMR methods as those used in this study. To further ascertain the high mannosylation of *C. reinhardtii*'s cell wall, our ssNMR approach could be applied on cell-wall extracts from which mannose residues would be removed by mannosidase and other enzymes[55]. Fluorescence imaging using mannose-specific antibodies followed by enzymatic treatment[56] could also be considered.

The presence of O-glycosylation in *C. reinhardtii*'s cell wall architecture, notably the O-glycosylation of Thr by Ara in HRGPs previously demonstrated, is confirmed by our MAS-DNP results[26,57,58]. In addition,

our GC-MS analysis of partially methylated alditol acetates (Fig. 2b) shows that Man linkages (1,2 and 1,3) are consistent with Man oligosaccharides attached to proteins via N-linked glycosylation[27]. Since Asn residues are abundant in LWGPs (*ca.* 15%), and can only be N-glycosylated - a post-translational modification never reported in *C. reinhardtii's* cell wall - at least some of the LWGPs GP4 to GP7 should be N-glycosylated via their Asn residues, with GlcNAc linked to Man-rich oligosaccharides[59].

Our results suggest a model in which glycans are spatially isolated from the amino acid residues, and we postulate the presence of abundant but short Man-rich oligosaccharides. We confirmed the O-glycosylation of Hyp (and possibly Ser) to Ara in HRGP, and propose two glycosylation bonds in LWGPs: an N-glycosylation of Asn with GlcNAc and oligo-mannoside, and an O-glycosylation of Thr to Man. Finally, we also report new intermolecular contacts between Hyp and Man that would constitute the interface between HRGPs and LWGPs.

## Glycans are more hydrated than proteins
Hydration heterogeneity in the cell wall is an essential feature for land plants[60]. This has been poorly investigated in microalgae and here we explored the water distribution in *C. reinhardtii*'s cell wall, using a water-edited 2D $^{13}$C-$^{13}$C DARR experiment (Supplementary Fig. 14)[61]. The results shown in Fig. 5a and Supplementary Table 15 indicate that water is more tightly bound to glycans than to amino acid residues. Man and Gal with low $C_1$ chemical shifts, probably corresponding to terminal units, are more hydrated than other glycans. This superior hydration of glycans correlates with faster MAS-DNP hyperpolarization

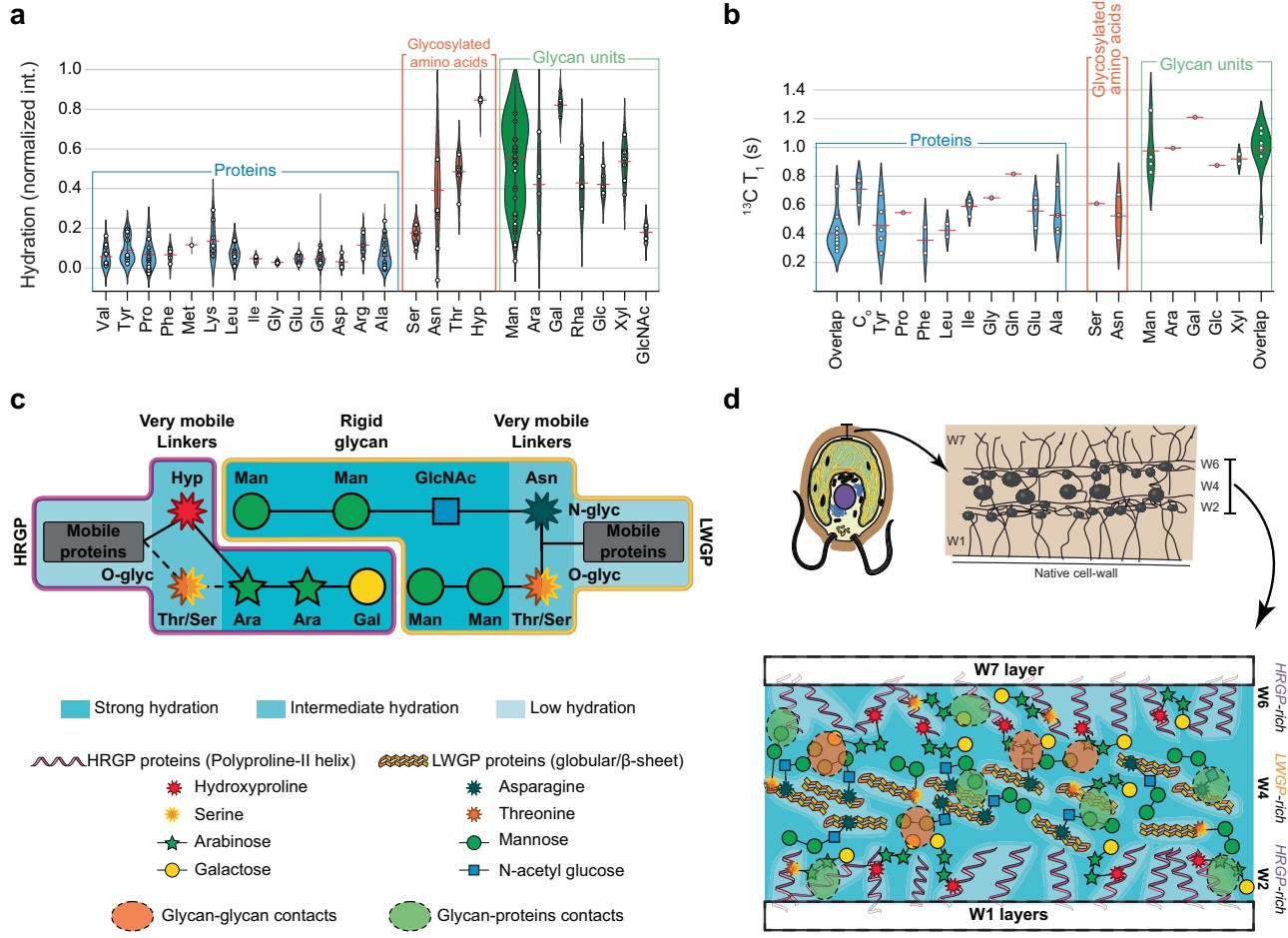

**Fig. 5 | Hydration, dynamics and detailed model of *C. reinhardtii*'s cell wall.**
**a** Molecular hydration of amino acids and carbohydrates (see experimental section and Supplementary Table 15 for details). **b** Dynamics assessment by [13]C spin-lattice (T1) relaxation analysis of amino acids and glycans. **c** Representation of the cell wall components as suggested by ssNMR and GC-MS results. HRGPs rich in Hyp are O-glycosylated with Ara and Gal. LWGPs are most likely N-glycosylated on Asn with Man-rich oligosaccharides and O-glycosylated on Thr. Man-rich oligosaccharides are probably at the very hydrated interface between those two protein families. Glycosylated amino acids intermediate hydration and dynamics is also represented.
**d** Schematic representation (top representation adapted from Goodenough et al.[19], used with permission of Rockefeller University Press, permission conveyed through

Copyright Clearance Center, Inc.) of the molecular network determined from ssNMR and MAS-DNP results obtained on natively hydrated cell walls with proteins being overall more dynamic than glycans. Key carbohydrates and amino acids are identified using the symbols in the legend. Hydration is color coded, dark blue regions corresponding to the more hydrated regions. W1 (inner layer, not extracted), W2 (fibrous internal polyproline II helical HRGP-rich), W4 (likely beta-sheet LWGP-rich, abundant in our sample), W6 (fibrous external HRGP-rich) and W7 (outer layer, not extracted) are tentatively represented to locate the results of our study in the overall microalgal extracellular matrix. Source data are provided as a Source Data file.

build-up (Supplementary information), which may indicate proximity with the biradicals dispersed in the water phase. Moreover, the residues that can be glycosylated, i.e. Hyp, Thr and, to a lesser extent, Asn and Ser, have a much higher hydration level than other amino acids, in some cases almost reaching the hydration level of glycans. However, GlcNAc are less hydrated, suggesting that they are involved in structural domains where water is either absent or weakly bound. This is compatible with their implication in N-glycosylation of Asn residues, which are also less hydrated than other glycosylated amino acids.

Considering the LC-MS results and the resolution of our MAS-DNP spectra, glycosylation of at least a fraction of Ser residues is highly probable. We explain the fact that Ser and Asn overall appear less affected by deglycosylation, and that their spatial contacts to glycans are hard to detect, from their dilution by pools of non-glycosylated amino acids.

As mentioned before, the oligosaccharides bound to amino acids in the cell wall extract should include an average of three sugar units. Moreover, we determined a glycan/amino acid ratio of 1:2, which

would correspond to 50% glycosylated amino acids with a single glycan or 16% with an average oligosaccharide length of three glycans. Therefore, Hyp and a significant part of Thr, Ser, and Asn residues must be glycosylated. As shown in Figs. 2c and 3a, our results are in good agreement with a model where all Hyp and approximately half of Ser are glycosylated in the form Hyp-O-Ara-Ara-Gal and Ser-O-Ara-Ara-Gal, as suggested in HRGPs[26], and potentially Thr-O-Ara-Ara-Gal. In addition, most remaining Thr and Ser residues as well as approximately half of Asn would be glycosylated in the forms Thr-O-Man-Man, Ser-O-Man-Man and Asn-N-GlcNAc-Man-Man in the LWGPs. There would be additional minor forms of glycosylated amino acids with Glc, Rha or Xyl glycans, but also possibly shorter, longer or branched oligosaccharides, as long as the ratio between glycans would remain similar[23,27].

Altogether, these results help refining the macromolecular organization of the cell wall, in which "dry" globular protein cores would be surrounded by hydrated layers containing glycans (Fig. 5a). Both regions would be linked through glycosylation, implying that the

protein surfaces would contain glycosylated amino acids, mostly Hyp, Ser and maybe Thr in the HRGPs, Thr, Ser and Asn in the LWGP. The significant hydration of glycans is probably an integral part of the high plasticity of the cell wall, as observed in higher plants[62,63]. Moreover, water can provide the flexibility enabling the glycan interactions that are necessary for the bridging of structural proteins[64]. In higher plants, this is for example required to form nanostructures in secondary cell walls, or at the hemicellulose-cellulose interface[65].

## Rigid glycans *and* mobile proteins

We then probed the dynamic properties of each chemical site by recording 1D ssNMR spectra with different $^1$H-to-$^{13}$C polarization transfer schemes: Cross-Polarization (CP) - which enhances the sensitivity of the rigid molecular segments – and Insensitive Nuclei Enhancement by Polarization Transfer (INEPT)[32,66] for dynamic segments Despite the low resolution of the 1D spectra, especially in the carbohydrate region, 44 distinct carbon resonances could be identified, i.e., 20 from glycans and 24 from amino acids (Supplementary Fig. 16 and Supplementary Table 17). The CP build-up was faster for glycans, indicating a strong dipolar-coupling network, comparable to crystalline starch[35] (Supplementary Fig. 17). However, the CP decay was slower for glycans, indicating less motion on the millisecond regime. Glycans with low $C_1$ chemical shifts, i.e. involved in terminal groups, were particularly favored by INEPT. Terminal Man and Gal units are therefore both highly hydrated and very mobile as compared to other polysaccharides, suggesting their presence at the water-amino acid interface. Polarization transfer efficiency differences also explain why amino acids signals, including Hyp cross peaks, were preferentially detected by CP-DARR, or at low temperatures (100 K) (Supplementary Figs. 8 and 9). This is most likely due to their unfavorable dynamics or relaxation rates[67].

Motional differences were more quantitatively determined by measuring $^1$H and $^{13}$C longitudinal relaxation times in the laboratory ($T_1$) and the rotating frame ($T_{1\rho}$) (Supplementary Figs. 17 and 18). While $T_1$ probes motions with correlation times on the order of ns, $T_{1\rho}$ is sensitive to slower (ms) motions. Comparing relaxation times to published values measured in other contexts permits to refine our model. No differences were observed in the $^1$H $T_1$ values, indicating that $^1$H polarization is homogenized throughout the sample by spin diffusion. On the other hand, carbohydrate $^{13}$C $T_1$ as well as $^1$H and $^{13}$C $T_{1\rho}$ values were slightly longer than those of amino acids, showing that glycans are less mobile than proteins, both on the nanosecond and millisecond scales. To illustrate these clear discrepancies between glycans and proteins, $^{13}$C $T_1$s are shown in Fig. 5b as an example, while the other $T_1$ and $T_{1\rho}$ values are plotted in Supplementary Fig. 18. The response of carbohydrates and proteins to polarization transfer schemes indicates a higher mobility of glycans, in particular for those in terminal groups (see Supplementary Fig. 16 section for more details). Overall, $^{13}$C and $^1$H relaxation behaviors - both on the ns and ms timescales - are not indicative of large crystalline regions but more characteristic of a semi-mobile material akin to surface cellulose or hemicellulose.

In conclusion, proteins and glycans differ in both dynamics and hydration, with mobile protein domains, more rigid but hydrated glycan domains, and glycosylated proteins at the interface. These two families of molecules are spatially segregated, even if it is in small nanometer-scale regions that are chemically and dynamically heterogeneous, similarly to layered semi-crystalline polymers or plant cell walls[68].

## Discussion

This work allowed refining the layered model of *C. reinhardtii*'s cell wall, which was primarily known from EM. The structural insights provided here by both ssNMR and MAS-DNP on a complex sample comprising proteins and glycans highlight the limitations of targeted analysis and data mining prediction. By studying a hydrated cell wall in a near native-environment in comparison with intact cells, we were able to obtain unique structural and dynamic information on the sugar and proteins that compose this complex biological scaffold. These molecular-level data improve the current model of the glycoprotein-rich cell wall of *C. reinhardtii*, and allow reconciling this organization with the cell wall function of microalgae, and higher plants to a certain extent. We determined the composition and relative abundance of glycans and proteins and the presence of previously underestimated lower molecular weight glycoproteins, LWGPs, as well as the high Man content of these samples. Our data support the presence of short length oligosaccharides, reveal the presence of β-sheets, the existence of both O- and N-linked glycosylations including mannosylation of Ser and Thr, show a spatial segregation of glycans and proteins in regions with differing dynamics and hydration, and uncover specific molecular interactions between them. The relevance of our findings was reinforced through comparisons with whole microalgal cells.

The surprising low abundance of Hyp residues and high abundance of Man are most likely explained by the preservation of LWGPs in our mild extraction process, which also contained the structurally important HRGPs previously identified. LWGPs are shown to contain short Man-rich oligosaccharides either O-glycosylated to Ser and Thr, or connected to GlcNAc that would be N-glycosylated to Asn. This provides an additional cross-linking strategy contributing to the structural integrity of the cell wall[69]. Considering that most of the Hyp detected in our sample are located in the HRGPs (Fig. 3b), which mostly contain oligosaccharides made of Ara residues and Man as a terminal group[23,24,26], the identified dipolar contacts of Hyp with Man would most certainly be not covalent, but due to spatial proximity, suggesting that Man is at the interface between the LWGPs and the Hyp/Ara-containing HRGPs.

These results, schematized in Fig. 5c, suggest that partially structured and highly mobile protein-rich domains are bound to less mobile and highly hydrated glycan-rich regions. The binding occurs by glycosylation to short (2–5 units long) oligosaccharides, thus contributing to the cohesion of the protein scaffold. Hyp, Ser and possibly Thr ensure O-glycosylation of HRGPs by Ara and Gal, while LWGPs are mostly O- and N-glycosylated by Man through Thr, Ser and Asn. At a larger scale, considering that we extracted the cell wall central triplet observed by TEM (between W2 and W7, Fig. 5d, top), the spatial segregation between protein- and glycan-rich regions by oligosaccharides suggests an alternating layered structure (Fig. 5d, bottom) in which hygroscopic glycans contribute to the cell-wall resiliency and flexibility. This model evokes the protective and hydrated surface of macroalgae enabling them to resist chemical and mechanical stresses[70].

Molecular-level details of the microalgal cell wall organization and rigidity is determinant to exploit this high-value renewable biomass[71]. The methodology employed here using in situ and in cell ssNMR and MAS-DNP paves the way to investigate the responses of microalgae to external contaminants or other stresses, or cellular processes during cell growth. This work is a step towards a better understanding of glycoprotein cell walls in intact cells.

## Methods

### Cell growth

The wild-type *wt* (CC-124), flagella deficient *bald2* (CC-4478 mt$^+$, also named *bld-2*)[72], cell-wall deficient *cw15* (CC-4349 mt$^+$), and starchless *sta 6-1* (CC-5374 mt$^+$) strains were obtained from the Chlamydomonas Culture Collection, University of Minnesota (Chlamydomonas Resource Center, http://chlamycollection.org). Whole *bald2* cells and cell wall extracts were used, unless specified. Cells were grown in a standard Tris bicarbonate-phosphate (TBP) medium buffered with HCl to pH 7.3[73]. The TBP medium was inoculated with microalgae

previously kept on TAP-medium 1.6% agar plates. Autotrophic growth conditions were imposed using $N_2$ bubbling before sealing the flask. Then cells were grown asexually using continuous white light illumination (100 µmol photons·m$^{-2}$·s$^{-1}$) at 23 °C ± 1 °C with gentle orbital agitation (100 rpm). Typically, five days were required to reach the late exponential phase ($5 \times 10^6$ cell·mL$^{-1}$) for 3 L of medium in a 6 L flask. Cells were harvested after a $1500 \times g$ centrifugation for 10 min. For sexually differentiated cells, both mating types CC-124 ("mt+") and CC-125 ("mt−") were grown separately until the early exponential phase, which was followed by centrifugation, resuspension in $N_2$-depleted medium, and mixing. After flagella loss, differentiation and gametogenesis, mated cells undergo sexual reproduction and usually have a thicker cell wall.

### $^{13}$C, $^{15}$N isotope labeling

The growth medium was supplemented with either 1 g·L$^{-1}$ of NaH$^{13}$CO$_3$ or 400 mg·L$^{-1}$ of $^{15}$NH$_4$Cl sterilized using 0.22 µm filters immediately prior to cell culture.

### Cell viability

A fluorescence analysis was performed on a flow cytometer (BD Accuri™ C6) to test the cell viability after NMR experiments 1000 intact cells were acquired for each sample. This test measures the natural chlorophyll fluorescence and fluorescein diacetate (FDA)-derived fluorescence generation based on esterase activity level[74]. A total of 5 µL of a 1 mM FDA solution was added to 1 mL of cell solution at $2 \times 10^6$ cells·mL$^{-1}$ for 20 min prior to measurement. Cell viability was shown to be ~90% before whole-cell NMR experiments, and above 60 % after 2 days of experiments at 298 K.

### Cell wall extraction

To prevent cell wall extracts from contamination with flagella, we used the *bald2* strain, although similar results were obtained with the wild-type strain (Supplementary information). The cell wall was extracted using Goodenough and Heuser's protocol[14]. Briefly, cells were grown until the late exponential phase was reached (~$5 \times 10^6$ cells·mL$^{-1}$), then harvested using a 15 min centrifugation at $1500 \times g$ and 8 °C. The cell pellets were rinsed twice with a 1 mM phosphate buffer (pH 7.3) supplemented with 86 mM NaCl to mimic the ionic strength of the growth medium. The cell wall was extracted by diluting the pellets with a 2 M sodium perchlorate (NaClO$_4$) solution, to reach a final concentration of $6 \times 10^8$ cells·mL$^{-1}$. Cells were gently mixed by hand for 30 min before centrifugation at $2500 \times g$ for 30 min at 8 °C. An additional extraction step was performed on the cell pellets to obtain cells totally devoid of a cell wall (called "uncovered cells"). The combined supernatants containing the cell walls were centrifuged at $30,000 \times g$ for 40 min to remove any cellular material and debris from the solution. The solution (about 30 mL) was then dialyzed against 10 L of a 0.05% sodium azide solution in nanopure water under gentle agitation for 5 days. The dialysis solution was renewed twice a day. A precipitate progressively appeared in the dialysis tubing. The dialyzed solution was then lyophilized before being resuspended again in a minimal volume of NaClO$_4$. The cloudy solution was dialyzed again – an important step in the cell wall reconstruction. It can be noted that to obtain a cell wall extract that contains only HRGPs, several additional dialysis and freeze-drying steps are necessary[19], and lead to an insufficient amount of material for ssNMR analysis. In our case, after the second dialysis, the cell wall sample was lyophilized and rehydrated just before the ssNMR experiments.

For experiments on whole uncovered cells, the pellet obtained after NaClO$_4$ extractions was rinsed at least 5 times in a 1 mM phosphate buffer (pH 7.3) supplemented with a 86 mM NaCl solution. The cell pellet was used without further purification for ssNMR and GC-MS experiments.

### Chemical deglycosylation

Chemical deglycosylation was performed as reported elsewhere[42]. Briefly, 100 mg of dry cell wall were cooled to 0 °C for at least 30 min. The dry powder was then added to 7.5 mL of a cold 2:1 v/v mixture of trifluoromethanesulfonic acid and anisole. After complete dissolution of the cell wall powder, the mixture was stirred at 0 °C under $N_2$ bubbling for 3 h. The solution was then cooled in a methanol-dry ice bath (−15 °C) for a few minutes, and 15 mL of diethyl ether were added, followed by a slow dropwise addition of 15 mL of 50% aqueous pyridine (v/v %) solution. The solution was gently mixed and the ether phase was removed. Ether extraction was performed a total of three times, and the aqueous phase was dialyzed against nanopure water with sodium azide. The protein extract was freeze-dried and hydrated prior to use.

### Solid-state nuclear magnetic resonance

All 800 MHz (18.8 Tesla) experiments were conducted on a Bruker Avance III HD NMR spectrometer using a 3.2 mm triple resonance HCN probe at 10 °C, and at 13.5 kHz MAS to prevent rotational resonance and spinning sideband interferences. The typical radiofrequency field strength was 83 kHz for $^1$H decoupling, using the two-pulse phase-modulated (TPPM) scheme, with 5.7 µs for each pulse. The radiofrequency field strengths were also 83 kHz for $^1$H hard pulses and 50 kHz for $^1$H ramped spin-lock, 63 kHz for $^{13}$C hard pulses and spin-lock, 42 kHz for $^{15}$N hard pulses, 34 kHz for $^{15}$N spin-lock. The key acquisition parameters of all 1D and 2D experiments are summarized in Supplementary Table 20. See Supplementary Table 21 for details of the sample replicate compared by ssNMR. All assigned peaks will be deposited to the Complex Carbohydrate Magnetic Resonance Database (CCMRD)[34], and the Bruker Topspin dataset is freely available upon request.

We assess the protein/glycan ratio by using fully relaxed (30 s recycling delay) 1D direct polarization (DP) $^{13}$C ssNMR spectra, and comparing the integrals of the carbohydrate C$_1$ (integral from 92.0 to 111.0 ppm, see the spectral regions highlighted in purple in Fig. 1e and Supplementary Fig. 16) and the amino acid carbonyl resonances (integral from 165.0 to 185.0 ppm, gray region in Supplementary Fig. 16). We measured a carbohydrate/protein molar ratio of 1:3 (adjusted to 1:2 considering the amount of Asp and Glu with their carboxyl end group). A significant part of the cell wall is thus composed of glycans, in agreement with the literature[27] and SDS-PAGE (Fig. 2a). Non-destructive qualitative assessment of the glycan/protein ratio is made available using simple 1D $^{13}$C ssNMR, while it is challenging for colorimetric or MS approaches known to be destructive, especially for complex eukaryotic samples.

To probe polysaccharides with diverse mobility, four 1D $^{13}$C spectra were recorded using different schemes to create initial magnetization. $^{13}$C DP spectra were acquired using either a 30 s or a 2 s recycle delay for quantitative detection of all molecules or a selection of mobile molecules, respectively. Cross polarization (CP) was used with a 1 ms contact time for rigid molecules. Refocused Insensitive Nuclei Enhanced by Polarization Transfer (INEPT) experiments with two 1.72 ms delays followed by two 1.15 ms delays were used for mobile species. Interestingly and as shown on Fig. 1e, native cell spectra can be reconstructed by a weighted sum of those of the extracts and remainder after extraction (black spectra Fig. 1e) proving that the extraction protocol is lossless and that the cell wall accounts, here, for 38% of the overall intensity.

The spin-lattice (T$_1$) relaxation time was measured using inversion recovery for $^{13}$C, and by inversion recovery followed by CP for $^1$H. $^{13}$C and $^1$H rotating-frame spin-lattice relaxation (T$_{1\rho}$) were measured using both CP build-up and Lee-Goldberg cross polarization (LG-CP) at 298 K under 10 kHz MAS. The spin-lock field was 62.5 kHz for the $^1$H-T$_{1\rho}$ measurement. Relaxation data were fit using a single exponential decay function.

To assign glycan and amino acid signals, a 2D $^{13}$C DP refocused J-INADEQUATE spectrum was recorded using a 2 s recycle delay as reported in plant cell-wall[75]. $^{13}$C CP refocused J-INADEQUATE was also employed to detect rigid molecules, using CP parameters described above for 1D experiments. INADEQUATE delays during the polarization transfer were set to 2.4 ms. The double quantum INADEQUATE spectra were sheared[76] using the "ptilt1" command available on Bruker Top-Spin software to enable comparison with single quantum spectra. The resonance assignments of 37 sub-forms of carbohydrates are summarized and compared with literature values in Supplementary Tables 2 and 3.

We also assigned 339 peaks corresponding to 87 different spin systems in proteins, among 18 different amino acids. All of them were quantified using both HPLC and ssNMR with a method that accounts for overlapping peaks[32] (Fig. 3a and Supplementary Fig. 8, and Supplementary Tables 8–11). Chemical shift prediction according to secondary structure was provided by TALOS+[77] and CSI 3.0 (http://csi3.wishartlab.com/)[78]. Chemical shift differences between polyproline II (PPII) helices and random coils are still hard to detect using these tools, especially since PPII helices remain underexplored by NMR. In our sample for instance, Ile, Lys, Thr and Val would be located in β-strand environments, while Asx, Gly, Pro and Ser would either be in random coils or in PPII helices (Fig. 3a). The propensity of an amino acid to be included in a β-sheet has been shown to be more correlated to its neighboring residues than to its chemical nature[79] and, indeed, no correlation could be established between the hydrophobicity or charge of the residues and their β-sheet propensity here. In summary, while we can assert the presence of β-strands and exclude that of α-helices in our sample, we cannot rule out the presence of PPII helices previously reported in HRGPs[20,23,48]. Additional information is given in Supplementary Fig. 4 and Supplementary Table 1–5.

2D $^{13}$C-$^{15}$N NCa and NCaCx spectra were acquired using a 1.7 s relaxation delay, 600 μs $^{1}$H-$^{15}$N CP, 5 ms $^{15}$N-$^{13}$C CP, followed by 100 μs or 100 ms $^{13}$C-$^{13}$C DARR correlation, respectively. DARR and PDSD experiments were recorded on a Bruker Avance III-HD spectrometer operating at a frequency of 599.95 MHz for $^{1}$H and 150.87 MHz for $^{13}$C. A 3.2 mm Varian triple resonance MAS probe was used, with a typical spinning frequency of 15 kHz, and the temperature was kept at 278 K with a dedicated cooling cabinet to prevent sample degradation. Dipolar based DARR and PDSD experiments were acquired with spin diffusion periods from 5 ms to 500 ms, and from 10 ms to 1000 ms, respectively. Peak intensities were followed using CCPNMR software accordingly to peak assignment described in Supplementary information. 13C $T_{1\rho}$ filter is performed using a continuous 50 kHz Lee-Goldberg spin-lock on the $^{13}$C channel before DARR mixing and detection.

Water-edited $^{13}$C-$^{13}$C DARR spectra[80] were initiated with $^{1}$H excitation followed by a $^{1}$H-$T_2$ filter of 0.88 ms ×2 that eliminates 97% of overall signals but retains 80% of water magnetization. Then a 4-ms $^{1}$H mixing period for water-to-glycan/amino acid transfer allows the best discrimination between hydrated and less hydrated regions. A 0.25 ms $^{1}$H–$^{13}$C CP was used for $^{13}$C detection. A 50-ms DARR mixing period was used for both the water-edited spectrum and the control 2D spectrum showing full intensity. Intensities were measured for all resolved and assigned correlations, and the intensity ratio between the water-edited spectrum and the control spectrum ($S/S_0$) was quantified. The error bar was calculated following the formula $(\frac{S}{S_0})^* \sqrt{\frac{1}{(sino)_S^2} + \frac{1}{(sino)_{S_0}^2}}$, with sino being the signal-to-noise ratio provided by TopSpin for a given peak on a slice of the 2D spectrum corresponding to the correlation of interest. See Supplementary Table 15 for details of the hydration intensities.

## Magic angle spinning - dynamic nuclear polarization ssNMR
The sample is mixed with a biradical solution, which unpaired electrons have a 658-fold higher spin polarization than $^{1}$H nucleus under the same experimental conditions. The electron spin polarization is transferred to the nuclei under microwave irradiation as the sample is maintained at a cryogenic temperature. The sample contains a cryoprotectant to avoid ice formation and preserve potentially fragile structures. A 10 mM biradical stock solution of AMUPol[81] was freshly prepared using $d_8$-glycerol/D$_2$O/H$_2$O (60/30/10 v%), referred to as the "DNP juice". Then 50 μL of the DNP juice were added to ~30 mg of the $^{13}$C,$^{15}$N-labeled *C. reinhardtii* cell wall and mixed for 5–10 min. About 30 mg of well-hydrated sample was transferred into a 3.2-mm sapphire rotor. For whole-cell experiments, fully $^{13}$C-labeled microalgae were centrifuged, then rinsed twice with a buffer containing 86 mM NaCl, before being resuspended in the DNP juice and pelleted after a few minutes in the solution.

All experiments were performed on a 600 MHz/395 GHz MAS-DNP spectrometer equipped with a gyrotron microwave (μw) source[82,83]. The cathode currents of the gyrotron were 160 mA. All MAS-DNP spectra were measured using a 3.2-mm triple resonance HCN probe at 8 kHz MAS frequency. The temperature was ~ 95–98 K with the μw off and ~100–110 K with the μw on. Typically, a 45-fold enhancement factor of NMR sensitivity with and without μw irradiation was achieved (Supplementary Fig. 1). We measured relatively short build-up time constants, $T_B$ = 3–5 s, indicating a good mixing of the biradicals and biomolecules in these cellular samples. 2D correlation experiments were implemented with 50-ms DARR, 1.5-s PDSD, NCa, and 100-ms NCaCx experiments. The total acquisition time was 6 h for the DARR, 8 h for the PDSD and 2 h for the NC experiments. $^{13}$C chemical shifts were externally referenced to TMS using adamantane's CH$_2$ signal set to 38.48 ppm[84]. The key acquisition parameters of all MAS-DNP experiments are summarized in Supplementary Table 20.

## Cryogenic temperature ssNMR experiments
Low temperature experiments were performed at 98 K under the same conditions described for the MAS-DNP experiments.

## Denaturing gel electrophoresis (SDS-PAGE)
SDS-PAGE was performed using 4–20% polyacrylamide (total concentration of both the acrylamide and bis-acrylamide crosslinker) gradient Mini-PROTEAN TGX Precast protein gels, with a Mini-PROTEAN II electrophoresis cell. Prior to any treatment, the dry cell wall was resuspended at 5 mg·mL$^{-1}$ in nanopure water. Samples are then diluted 5 times with the proper amount of Laemmli buffer, reduced with β-mercaptoethanol (5 % final concentration), and treated for 25 min in an 85 °C bath. For electrophoresis, 10 μL corresponding to 10 μg of sample was injected per well, and Tris-glycine SDS running buffer (25 mM Tris, 192 mM glycine, 0.1% SDS, pH 8.3) was used at 10 mA for 15 min then 50 mA for approximately 1 h. The prestained protein ladder Precision Plus Protein Kaleidoscope for proteins in the molecular range 10–250 kDa was used as protein standards. After electrophoresis, total protein staining for relative protein proportion evaluation was performed using Coomassie Blue (0.05% w/v Coomassie Blue R-250) for 40 min, and destained with 30% v/v methanol/ 10% v/v glacial acetic acid for 5 h. Gels were acquired and visualized using ChemiDoc MP and ImageLab. Final image editing and protein quantification was performed with the same software. Error bars correspond to the standard deviation obtained after quantifying two cell wall extracts per strain. Uncropped pictures of the gels are found in the Supplementary Fig. 19.

## High-performance liquid chromatography amino acid analysis
Samples were vacuum dried, suspended in 6N HCl containing 1% phenol, and hydrolyzed for 24 h at 110 °C under N$_2$ stream. Following hydrolysis, samples were vacuum dried to remove excess HCl, resuspended in a methanol:water:trimethylamine (2:2:1) redrying solution and dried under vacuum for 15 min. Samples were derivatized for 20 min at room temperature in methanol:water:trimethylamine:phenylisothiocyanate

(7:1:1:1) solution. This solution was then removed under vacuum, and the samples were washed with the redrying solution and vacuum dried for an additional 15 min. Chromatographic analysis was performed using a Waters Acquity Ultra-High Performance Liquid Chromatography (UPLC) System. Samples were dissolved and injected into an ethylene bridged hybrid C18 column, and a modified Pico-Tag gradient was used at 48 °C, with phenylthiohydantoin (PTH)-derivatized amino acid detection occurring at 254 nm. Cysteines were protected by performing acidic oxidation prior to hydrolysis. HPLC data correspond to the average of 4 replicates (Supplementary Table 11).

### Glycosyl composition and linkage analysis by GC-MS

Glycosyl composition analysis of neutral sugars was achieved after an acetone extraction on dried cells removing most pigments in the sample. The dried sample was then dispersed in 0.5 mL of 2 M trifluoroacetic acid (TFA) in a sealed reaction tube. After 20 min of sonication in an ultra-sound water bath at room temperature, hydrolysis was performed at 121 °C for 2 h, followed by overnight reduction with $NaBD_4$, and 1 h acetylation with acetic anhydride and pyridine (1:1, v/v) at 80 °C. To obtain glycosyl linkage information, the cell wall is suspended in DMSO and, after permethylation, hydrolysis, reduction and acetylation, it is converted into partially methylated alditol acetates (PMMAs) that can be efficiently detected by GC-MS[85].

GC-MS analysis was performed on an HP-5890 GC interfaced to a mass selective detector 5970 MSD using a SupelcoSP2330 capillary column (30 × 0.25 mm ID) with the following temperature program: 60 °C for 1 min, then ramp to 170 °C at 27.5 °C·min$^{-1}$, and to 235 °C at 4 °C/min, with 2 min hold, and finally to 240 °C at 3 °C·min$^{-1}$ with 12 min hold. Inositol was used as an internal standard. Glycosyl constituents were assigned, based on the GC-MS retention time of the derivatives of sugar standards, and on electron ionization MS (EI-MS) fragments of $^{13}$C-labeled alditol acetate derivatives. Results of the quantitative GC-MS glycan composition is given in Supplementary Table 1 and identified glycan linkages are displayed in Supplementary Table 5. The relative EI detector response (percentage) obtained by GC-MS is a semi-quantitative reporter of linkage proportion, and is provided in Supplementary Table 1 for several samples, including whole cells.

### O-glycoproteomic analysis by bottom-up mass spectrometry

Proteins from *C. reinhardtii's* cell wall were further extracted in 50 mM ammonium bicarbonate, 1% Rapigest before trypsin digestion, PNGase F digestion and Concanavalin A lectin weak affinity chromatography (ConA-LWAC) and mass spectrometry[51,86]. Our data mining strategy included variable hexose modifications of Ser or Thr residues for identification of Hex1, Hex2 or Hex3 modifications at each site, and we found evidence suggesting O-linked glycosylation of Ser and Thr with Hex1 and Hex2 modifications (Supplementary Fig. 20).

### Scanning electron microscopy

Cells were first harvested by centrifugation, washed with deionized water, and fixed in 2.5% glutaraldehyde in 0.1 M cacodylate buffer (pH 7.2) for 2 h at 4 °C, and again in 1% buffered osmium tetroxide for 1 h at 4 °C. Samples were then washed three times with 0.1 M cacodylate buffer. Dehydration was carried out with a series of ethanol washes with concentrations from 10% to 100% on round glass slide. Specimens were coated (5 nm thickness) using a CCU-010 HV (turbo-pumped) Safematic Coating System. The SEM analysis was carried out with a high-resolution system capable of operating of up to 200 kV, equipped with an EDAX Genesis EDS on Philips CM200 and AMTXR-60BCCD camera.

### Transmission electron microscopy

Microalgae were harvested by centrifugation, then washed several times with sterile isotonic buffer prior to cell fixation as describe above. Specimens were infiltrated with a resin, and thin sections of 70 nm were obtained using a diamond knife on a Reichert-Ultracut ultramicrotome. The sections were mounted on copper grids and coated with carbon for TEM observation. Images were recorded using a FEI Tecnai 12 BioTwin microscope operating at 120 kV and equipped with an AMT XR80C CCD camera system.

### Supplementary Information

The online version contains supplementary material available at https://doi.org/xxxxxx. Cell-wall extract and control samples additional 1D/2D MAS-DNP and ssNMR spectra, complete glycan (by ssNMR and GC-MS) and protein (by ssNMR and HPLC) peak assignments and integrations, glycan linkages composition determined by GC-MS, mannosylated protein sequences identified by LC-MS, polarization build-ups (MAS-DNP, CP and PDSD), supplementary relaxation charts, detailed hydration calculation and a list of experiments with corresponding acquisition parameters can be found in the supplementary information.

### Reporting summary

Further information on research design is available in the Nature Portfolio Reporting Summary linked to this article.

## Data availability

Source data are provided with this paper. The NMR data generated in this study have been deposited in the Zenodo public database under accession code 10403721. The data generated in this study are provided in the Supplementary Information/Source Data file. Source data are provided with this paper. All relevant data that support the findings of this study are available within the article, as well as in the supplementary information and source data tables. The source data underlying Supplementary Tables ("Tables_Poulhazan_20231219" Excel file), Figs. 2d, 3a, b, and 5a, b and Supplementary Figs. 10, 15, 17, 18, and 12c ("SourceData_Poulhazan_20231219" Excel file) are provided as a Source Data file. The mass spectrometry proteomics data have been deposited to the ProteomeXchange Consortium via the PRIDE[87] partner repository with the dataset identifier PXD048024. Source data are provided with this paper.

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

## Acknowledgements

This research was funded by the Natural Sciences and Engineering Research Council (NSERC) of Canada (grant RGPIN-2018–06200 to I.M.), the Centre National de la Recherche Scientifique (UMR 7203 to D.E.W.), and the U.S. Department of Energy, Office of Science, Basic Energy Sciences (grant DE-SC0023702 to T.W.). A.P. would like to thank the *Fonds de recherche du Québec – Nature et technologies* (FRQNT), and the Quebec Network for Research on Protein Function, Engineering, and Applications (PROTEO) for the award of scholarships. The glycosyl composition and linkage analyses were supported in part by the Chemical Sciences, Geosciences and Biosciences Division, Office of Basic Energy Sciences, U.S. Department of Energy grant (DE-SC0015662) to P.A. at the Complex Carbohydrate Research Center. The National High Magnetic Field Laboratory is supported by National Science Foundation through NSF/DMR-1644779 and the State of Florida, the MAS-DNP instrument is supported by the NIH P41 GM122698 and NIH S10 OD018519. I.M. is a member of PROTEO and Resources Aquatiques Québec. A.H., S.Y.V., and H.J.J. wish to acknowledge funding from Danish National Research Foundation grant DNRF107, Novo Nordisk

Foundation grant NNF22OC0076899, Mizutani Foundation for Glycoscience and VILLUM FONDEN grant 00025438. We would like to thank S. Waffenschmidt (University of Cologne, Germany), U. Goodenough (Washington University in St. Louis, USA), O. Vallon and F. Zito (IBPC, Paris, France), B. Henrissat (AFMB, Marseille, France), and H. Wandall (University of Copenhagen, Denmark) for enlightening discussions.

## Author contributions

A.P., A.A.A., T.W., D.E.W. and F.M.V. designed and conducted the NMR and MAS-DNP experiments. A.P. and F.M.V. prepared and optimized the DNP samples. A.M. and P.A. conducted the GC-MS experiments. A.H., S.Y.V. and H.J.J. conducted the LC-MS experiments. A.P., A.M. and F.M.V. analyzed the experimental data. A.P., A.A.A., T.W., D.E.W. and I.M. wrote the manuscript.

## Competing interests

The authors declare no competing interests.
