## [Peer Review File · Nature Communications]

Molecular-Level Architecture of Chlamydomonas reinhardtii's Glycoprotein-Rich Cell WallREVIEWER COMMENTS

Reviewer #1 (Remarks to the Author):

Poulhazan et al. presented extensive characterization of algal cell wall extracts by solid-state NMR, revealing their molecular composition, interaction, and dynamics. Interpreting the results, the authors offered a refined and potentially controversial view of *Chlamydomonas* cell wall composition and architecture. On the other hand, the presence of N-glycosylated proteins in the *Chlamydomonas* cell wall is no surprise.

Overall, the manuscript is difficult to comprehend due to the insufficient description of the ssNMR results. Although I do not list here exactly which parts of the figures are not explained well nor mentioned, I want to share the expected difficulty of general readers in the Nature communications journal. The difficulty is exacerbated by placing critical information in the supplemental results again without sufficient explanation.

Below, I ask the authors to address three major concerns in the manuscript.

1. The authors claimed that the results describe the non-crystalline and insoluble part of cell wall in the abstract. The cell wall extracts were prepared by the Goodenough and Heuser's protocol extracting the perchlorate-soluble part of the cell wall. The insoluble part of the cell wall is defined as those inextractable by chaotropic salts such as perchlorate. Therefore, the authors analyzed the extracts of the highly 'soluble' part of the cell wall.^{[1][1][1]}_{SEP1SEP1}

The authors emphasized 'mild extraction' as a way to include non-HYP LWPs in their cell wall extracts. According to the described extraction method, 'mild' is based on omitting multiple purification steps to enrich crystalline HYP-glycoproteins. Therefore, the 'mild' extract may contain some non-crystalline parts of cell wall proteins, but the perchlorate-insoluble parts are still non-extractable by the given method.^{[1][1][1]}_{SEP1SEP1}

After the extraction with perchlorate, the W1/W2/W7 layers remain as insoluble on the cell. Thereby, the authors mentioned that their extracts did not contain W1, W2, and W7 layers in the top paragraph of page 5. So, the question is which parts of the cell wall the authors' extract represents. The answer is critical for how the results should be interpreted. For instance, the reported LWGPs may be contaminants that adhere to the other perchlorate-soluble components of the cell wall, such as secreted proteins associated with the W4 and W6 layers.^[1]_{SEP1}

2. Another concern is the lack of statistical evaluation of the quantitative data in this manuscript. Biological replicates are necessary for any scientific presentation of quantitative data. However, it is rarely found in the manuscript's data. The HPLC-based cell wall amino acid profile in Table 2 and the supplementary table 10 are among the exceptions. Furthermore, the method section does not mention how many biological replicates have been analyzed.

3. Of the intermolecular interactions described in this manuscript, two claims are either contradictory to previous literature or not expected in the cell walls of green plants. Therefore, these claims need to be further elaborated in the manuscript before making any conclusion.

The first one is O-glycosylated threonine. Based on Fig. 3c showing that Threonines interact mainly with Mannose, should we consider whether Man-O-Thr is present in the Chlamydomonas cell wall? First, I suggest the authors explain the evidence supporting the identification of O-glycosylated threonine in the main text using one of the main figures or tables. Second, the authors must thoroughly consider other information on whether to support the O-glycosylated threonine and which sugar is linked to threonines in the Chlamydomonas cell wall. Also, I want to see the authors explain that the observed O-glycosylated threonines are unlikely from a contaminated source material included in the final cell wall extracts.

The second one is the lack of O-glycosylated serine. There is no serine signal indicated in Figure 3c. However, glucose or rhamnose and mannose/galactose show connectivities to unknown amino acids (first paragraph on page 13). So, the unknown amino acids may include serine. If the authors disagree, please explain why serine is an unlikely candidate for the unidentified amino acids close to galactose.

Curiously, in the last paragraph of page 13, Hyp, Thr, and Asn backbone nitrogens were commented as connected amino acids to glycans. However, Supplemental figure 8 shows that Hyp/Thr/Ser signals are clustered as if they are indistinguishable. If Serine signals are not found in any dipolar coupling (C-C) or 15N-13C connectivity analysis, please explain how the authors concluded that serine is devoid of glycosylation.

It is critical to carefully address the absence of O-glycosylated serine in the data. Because this is contradictory to Gal-O-serines characterized in the plant cell wall and the serine-specific galactosyl transferase identified from the Chlamydomonas cell wall extract (Saito et al., Identification of Novel Peptidyl Serine-Galactosyltransferase Gene Family in Plants. JBC 2014).

Reviewer #2 (Remarks to the Author):

The manuscript by Poulhazan et al. describes solid-state NMR experiments, complemented with mass spectrometry analysis, on the cell wall of *C. reinhardtii*, which is a model microorganism for microalgae.

The authors made an attempt to describe, at atomic-resolution, the cell wall organization of this model system. However the study falls short in providing new and exciting results compared to the JACS paper published by the same authors in 2021 (Identification and Quantification of Glycans in Whole Cells: Architecture of Microalgal Polysaccharides Described by Solid-State Nuclear Magnetic Resonance, DOI: 10.1021/jacs.1c07429). Although the JACS 2021 paper provided new insights on how solid-state NMR experiments could provide meaningful details to scrutinize algae cell wall, a very similar analytical methodology is used in the present manuscript, again on a model algae system, therefore the study clearly lacks novelty. The composition of the cell wall of this system was already known. It is also unclear how the NMR analysis could help to understand the cell wall life cycle of such algae, e.g. the so-called "cell wall regeneration" would have been an interesting mechanism to explore.

The manuscript is well executed from a NMR point of view, and would be more suitable for a NMR-oriented journal or a journal focused on analytical (bio)chemistry.

Specific comment: the last sentence of the abstract highlights that the proposed methodology could be "transposable to study other microorganisms and plant materials", however this solid-state NMR methodology has been already used many times for fungal micro-organisms by one of the corresponding authors (T. Wang), not mentioning all the work from his previous supervisor (Mei Hong) on plant cell wall using NMR. This sentence should be changed.

Reviewer #3 (Remarks to the Author):

Comments to the authors

In the submitted manuscript entitled "Molecular-Level Architecture of *Chlamydomonas reinhardtii*'s Glycoprotein-Rich Cell Wall" by Poulhazan et al., the authors used solid-state NMR as the main tool to understand the organization of the extracted glycoprotein-rich cell wall of *Chlamydomonas reinhardtii* by probing the protein and glycan composition, spatial contacts, hydration state and dynamics. Authors show that the low molecular weight glycoproteins (LWGPs) make ~80% of total protein of the extracted cell wall when using a mild extraction protocol. They also report the high content of mannose in the extracted cell wall. The author then demonstrate that mannose-rich short oligosaccharides play an important role at cross-linking interface of the N-glycosylated LWGP domain and the O-glycosylated

HRGPs domain. In addition, the authors show a clear differentiation between the less hydrated and more mobile protein domain, the more hydrated and less mobile glycan domain, and the glycosylated protein domain with intermediate hydration state and mobility in the cell wall. Together with these findings, authors suggest a refined model of the molecular network for the cell wall of *C. reinhardtii*.

This work adds important information to the current understanding of the *C. reinhardtii* cell wall, and is certainly of interest to the cell wall and microalgae community. The paper is well written. The approach and methods used are well developed in the field, and the experiments are well designed. The data presentation is high quality and clear and linked to the conclusions.

However, my major concern is that the authors primarily investigated the architecture by reconstituting cell wall extracts. It is not clear how representative this is of the native cell wall architecture/organization. Despite the 1D and 2D PDSM ssNMR (Fig. 1e & 1f) showing similar peaks between whole cell and cell wall extracts, the extracted cell wall components may not reorganize as if they were in the native form, and this may not be reflected by the (Fig. 1e & 1f) comparison. In particular, the PDSM experiments were later used for glycan-protein contact analysis and peak intensities were used for quantification. Authors need to better clarify in the discussion how they ensure that the extracted cell wall architecture represents their native form. Maybe inclusion of a comparison between the difference spectra of “whole cell” and “whole cell cw-extracted” vs the “cell wall extract” spectra can help.

Another suggestion to the authors is that they need to provide more background information on a few topics/concepts that are discussed in the results section, which can greatly help the audience follow the story. Technical terms were not introduced but brought up in later results section, which make the story hard to follow. For example, introduction to the current understanding of the composition, structure and organization of *C. reinhardtii* glycoprotein-rich cell wall. More introduction/definition of the HRGPs, and low molecular weight glycoprotein (14-3-3) is needed, including brief descriptions of core structures, and what is the current understanding of their functional roles in the algal/plant cell wall. Introduction of N- and O-linked glycosylation would also be beneficial.

Other comments:

1. Some statements need references to support them e.g “that of *C. reinhardtii* is built on glycoprotein-rich layers that can be compared to extensins in higher plants”.
2. Fig. 1e is hard to understand. In the text (line 86 and the SI Fig.1 caption), authors described MAS-DNP is required cryogenic temperature, but why collected the spectra separately for MAS-DNP and cryogenic temperature? It seems the figure caption is wrong. Blue spectra are MAS-DNP with cryogenic temp., and the orange spectra are collected without MAS-DNP. Authors need to clarify this, and clearly label the spectra.
3. Line 94 and Fig. 1g, authors need to provide more description on the cell-wall deficient (cw15) strain, and explain in the text how it should be compared to the “whole cell cw-extracted” spectrum.
4. Line 115-118, authors didn't specify how the SDS-PAGE indicate similar glycosylation level (Fig. 2b) as results indicated by ssNMR (Fig.2a). The SDS-PAGE results discussed in the earlier two sentences don't provide direct link to conclude the last sentence of the paragraph. As the authors state in Line 218, the caption of Fig. 2b states “lane 6 : deglycosylated 13C-labeled bald2 cell wall with tentative molecular

weight assignment;" it seems unclear how much glycan was removed. The Authors need to clarify why they conclude that the SDS-PAGE indicate similar glycosylation level (Fig. 2b) as results indicated by ssNMR (Fig.2a), or these areas in the text need to be revised

5. In Fig 2b caption, authors mentioned the bald2 strain was used for the ssNMR study and more details were not provided until the method section. This needs to be brought up to early text in the results section or the introduction. And description of this strain and why using this strain for the study needs to be discussed.

6. In the sentence from Line 119-124 and the caption of Fig 2c, authors need to specify that the "extracted cell wall" is analyzed.

7. Line 131, typo in "groups of oligosaccharides (blue bars on Fig. 2e)". There are no blue bars in Fig. 2e

8. Even though the authors specified the low percentage of HRGPs (20%) of total protein content, considering a third of HRGPs should be Hyp, as the authors also stated in the text, detected Hyp is surprisingly low (~1%). Can the authors provide more explanation on the low abundance of Hyp detected by HPLC and ssNMR?

9. Fig. 3a and SI Fig. 6, spectra legend should be corrected to "extracted cell wall" vs "deglycosylated extracted cell wall". Also, for Fig. 3a-c, spectra axis labeling is covering the tick marks.

10. Sentence in line 248-250, "Glycans thus appear to have ... proteinated content of the cell wall" is hard to parse, and would benefit from rephrasing.

11. Line 295, "allow determining" to "allow to determine"

12. Line 299, "Fig. 3d and Supplementary Fig. 7" should be "Fig. 3d and Supplementary Fig. 8"

13. Line 309, "long oligosaccharides" should be changed to "long polysaccharides" to be consistent with later text.

14. Line 350-352, sentence, "Moreover, the residues that can be glycosylated, namely Hyp, Thr and, to a lesser extent, Asn and Ser, are hydrated to much higher levels close to those of glycans'", seems reversed.

15. Line 397, "...detected by CP-PDSD, or at low temperatures (< -160°C) (Supplementary Fig. 2)" seems wrong figure indicated. Authors meant the Fig.3b&c in the main text?

16. Line 452, typo "macroalgae"

Responses to the reviewers

We would like to thank the Reviewers for their careful reading of the manuscript and for providing insightful and helpful advice on cell wall biochemistry and NMR spectroscopy. Their recommendations enabled us to improve the clarity and readability of the manuscript, and gave us the opportunity to perform new experiments and to gain additional information from our results. The answers to all the questions are found below. All changes in the manuscript are highlighted in its revised version. Because these modifications have significantly increased the length of the manuscript, we had to shorten some paragraphs, and move some information to the Supplementary Information section. We hope this revision answers the questions satisfactorily and that our revised manuscript is fit for publication.

Reviewer 1

We would like to thank Reviewer 1 for the insightful comments, which enabled us to provide stronger evidence to our hypotheses and modify the manuscript to improve this aspect. The Reviewer's questions are also related to the novelty of our findings, which in some cases contrast with the existent literature in the field. We hope that the Reviewer will find the new version of the manuscript convincing.

1) The authors claimed that the results describe the non-crystalline and insoluble part of cell wall in the abstract. The cell wall extracts were prepared by the Goodenough and Heuser's protocol extracting the perchlorate-soluble part of the cell wall. The insoluble part of the cell wall is defined as those inextractable by chaotropic salts such as perchlorate. Therefore, the authors analyzed the extracts of the highly 'soluble' part of the cell wall.

The authors emphasized 'mild extraction' as a way to include non-HYP LWPs in their cell wall extracts. According to the described extraction method, 'mild' is based on omitting multiple purification steps to enrich crystalline HYP-glycoproteins. Therefore, the 'mild' extract may contain some non-crystalline parts of cell wall proteins, but the perchlorate-insoluble parts are still non-extractable by the given method.

*After the extraction with perchlorate, the W1/W2/W7 layers remain as insoluble on the cell. Thereby, the authors mentioned that their extracts did not contain W1, W2, and W7 layers in the top paragraph of **page 5**. So, the question is which parts of the cell wall the authors' extract represents. The answer is critical for how the results should be interpreted. For instance, the reported LWGPs may be contaminants that adhere to the other perchlorate-soluble components of the cell wall, such as secreted proteins associated with the W4 and W6 layers.*

We thank the reviewer for this very important comment that led to significant changes in the manuscript. Indeed, adequately defining the cell-wall extract that we are studying here is critical. We are aware of the many protocols that exist for plant cell-wall extraction, but as the Reviewer mentions, there is a consensus for considering Goodenough and coworkers' (Goodenough *et al.* 1986, Goodenough *et al.* 1988) as the most reliable in the case of *C. reinhardtii* using a chaotropic agent (sodium perchlorate here), as reported even before by Hills *et al.* (1975). In this context, the first point that Reviewer 1 raises reflects a terminology problem long persisting in the literature, as several authors have historically called this extracted cell wall "insoluble", because

it is insoluble in water. However, the Reviewer is right: the extracts should be more precisely described as “non-crystalline water-insoluble”. As pointed out by Reviewer 1, and as stated in our manuscript, we mostly extract the “central triplet” made of layers W4 and W6, leaving behind most of the “really insoluble” layers W1, W2 and W7. We now specify the water-insolubility but perchlorate solubility throughout the manuscript, notably in the Abstract and Introduction sections (**pages 2-4**). We also specify that the central triplet is composed of the W4/W6/W4 layers in the Introduction on **page 3**. The central triplet identification (as composed of layers W4 and W6) was already indicated in **Figure 1a**, but we have highlighted this information in this figure.

One key finding of our work is the discovery of unreported LWGPs associated to the cell wall, which we believe are not contaminants. This is a crucial point that we have carefully addressed. The presence of such proteins in our cell wall extracts had already been reported in the literature, suggesting that LWGPs are actual constituents of the cell wall. Already in 1975, Jiang and Barber (1975) detected a water-soluble fraction of *C. reinhardtii*'s cell wall that was rich in mannose. In the early 2000's, small cell-wall associated glycoproteins, such as the “14-3-3 proteins”, were also observed by Voigt *et al.* (2001). These glycoproteins were proved to be rich in serine and threonine residues, and were also reported as being more water-soluble than HRGPs. It is reasonable to presume that these soluble fractions would have been slowly removed by the sequential dialyses performed by Goodenough and coworkers (Goodenough *et al.* 1986, Goodenough *et al.* 1988), while they were retained by our adapted protocol. Indeed, Goodenough's protocol was mostly devised for Scanning Electron Microscopy (SEM) experiments, for which a very pure, dilute and homogeneous, almost recrystallized, sample was necessary. Typically, such a protocol preserves the largest objects. In our case, the whole mixture was kept and studied “as is”, including small constituents such as LWGPs.

We have several arguments in favor of the native state of our reconstituted cell wall that we hope will convince Reviewer 1, as well as the readers:

- (1) Microalgae culture contamination by any heterotrophic organisms such as bacteria, yeasts or fungi is highly improbable considering that microalgae were grown in photoautotrophic conditions with only bicarbonate as carbon source. All glassware was, of course, systematically autoclaved before use. We could not detect macroscopic signs of contamination in any of our samples, which appeared as characteristic homogeneous green pellets. Contamination by other photosynthetic organisms can also be excluded considering the sterile conditions used, and the fact that no other photosynthetic organisms are grown in our laboratory. Most importantly, no contamination was found by confocal, flow cytometry or electron microscopy in any of the numerous replicates throughout the study.
- (2) Contamination during the extraction process can be ruled out since the compositional analysis of the extracts and cells after extraction appear to be reproducible and without loss within experimental error. This is concluded from **Tables S6-S7**, which display the GC/MS results of the glycan content of cells before and after wall extraction, as well as in the extracted cell wall. The fact that we find a similar proportionality factor for each glycan, confirms the quality of our reconstitution protocol: practically no sugar unit is lost or altered during the extraction and reconstitution. This result is further supported by NMR

analysis in the new **Figure 1e**, which shows the almost exact superposition of the intact cell spectrum and the sum of the spectra from the extracted cell wall and whole cell after cell wall extraction. There is thus no loss or addition of components during the extraction protocol. Some variability in the glucose content can be observed, but it is mostly associated with the variability in the starch metabolism among sample replicates.

- (3) Wild-type (*wt*) cells are composed of 12.9% mannose (28.3% excluding glucose) whereas the cell-wall deficient *cw15* strain is composed of 2.4% mannose (16.1% excluding glucose). Therefore, the *cw15* strain contains a significantly lower mannose concentration as compared to the *wt* cells - a good indication that mannose is involved in the cell wall.

¹³ C alditol	bald2 whole cell	bald2 cw extracte d	bald2 cw extra ct	wt whole cell	wt cw extra ct	wt cw extra ct	cw15 whole cell	cw15 cw extra ct	sta6-1 whole cell	sta6-1 cw extra ct
Man	35.2	20.0	51.2	28.3	14.5	49.4	16.1	30.9	29.7	13.6
Ara	29.6	38.7	20.4	29.2	34.9	20.0	63.9	30.0	25.9	32.4
Gal	20.6	22.2	19.0	25.4	29.0	20.4	11.6	22.6	31.0	36.0
Rha	6.3	10.1	2.2	2.0	1.8	2.7	0.0	2.1	3.3	5.4
Xyl	2.2	3.0	1.5	3.5	5.0	1.5	6.8	4.7	2.3	5.0
GlcN Ac	6.1	6.2	5.7	11.4	14.8	6.0	1.4	9.7	7.8	7.5

- (4) We have recorded MAS-DNP spectra of whole microalgae on which the native cell wall is visible. Despite limited resolution, we can observe the same cross peaks as those seen with the cell wall extract, notably involving mannose. Since the radical used to obtain MAS-DNP spectra is unable to cross the cell membrane, the mannosylated glycoproteins that we observe in our extracts are therefore already present in the native cell wall environment.
- (5) Although this was not clearly stated in our manuscript, we have performed several replicates as well as the extraction protocol numerous times (18 biological replicates for strains containing cell walls), and on several strains (\pm flagella, \pm starch metabolism, or with different mating types, see table below), and we have reproducibly found the same additional glycoproteins, and the same overabundance of mannose, on SDS-PAGE gels, NMR spectra, and MS results. All these replicates have been studied at least by ssNMR, probing their overall molecular composition. The different strains that were analyzed and the number of replicates for each strain are summarized in the new **Table S21** in the Supplementary Material (and below for the Reviewer).

Strain used for cell-wall extracts	Number of biological replicates
Wild-type (CC-124)	6
Wild-type - deglycosylated	2
Flagella deficient bald2	6
Flagella deficient bald2 - deglycosylated	2
Cell-wall deficient cw15	2 (no cw extract)
Mating types CC-124 (“mt+”)	2
Mating types CC-125 (“mt-”)	2
Starchless sta 6-1	2

- (6) Significantly, almost no material and therefore no mannose-rich proteins are recovered when the extraction protocol is performed on the *cw15* strain devoid of a cell wall.

Altogether, we believe that points 1 to 6 are solid proofs that the mannose-rich proteins observed in our reconstituted cell walls do not come from external contamination. They are systematically present in reproducible quantities in our extracts for multiple replicates.

Although our results support a high mannose content in the cell wall, let's consider the various *Chlamydomonas* mannose-rich proteins reported in the literature and their potential involvement with our cell wall extracts. The presence of mannosylated glycoproteins in *Chlamydomonas* has been reported by Jiang and Barber, Vogeler *et al.*, Mamedov and Yusibov, and Mathieu-Rivet *et al.* (Jiang *et al.* 1975, Vogeler *et al.* 1990, Mamedov *et al.* 2011, Mathieu-Rivet *et al.* 2020). Mathieu-Rivet *et al.* described them as being mostly secreted and some are being described as periplasmic and suggested to be involved in “nutrient acquisition, cell–cell recognition, or cell wall degradation”. For all strains used in our study, small water-soluble proteins secreted in the medium by the microalgae can definitely be excluded because they would disappear during the numerous rinsing and dialysis steps. In the case of larger mannose-rich proteins that would not be eliminated by dialysis, they would have been recovered when the extraction protocol was performed on the cell-wall deprived *cw15* strain, but this was not the case. It is possible that these proteins remain attached to the remainder of the cell wall (layers W1-W2-W7) in the *cw15* strain, but this part of the cell wall is not the object of the present study as specified in the previous point.

Reviewer 1 suggests a second hypothesis: that these LMW mannose-rich proteins originate from the microalgae without being an integral part of the native cell wall. They would strongly interact with the extractable cell wall components from the W4-W6 layers without being part of these layers in the cell context, even in sodium perchlorate, and would not be efficiently eliminated by dialysis. Admittedly, this hypothesis cannot be excluded, and refers to the “non-canonical cell-wall proteins that can be considered as contaminant proteins” in *A. thaliana*, as discussed by the group of Jamet (Jamet *et al.* 2008, San Clemente *et al.* 2022). As regards to this statement, it is reasonable to consider that if a protein strongly interacts with cell wall components - sufficiently to go through the extraction protocol, two dialysis steps, renaturing non-chaotropic conditions and centrifugation - it is either as part of the cell wall, or strongly associated with it.

The strong interactions of the LMW mannose-rich proteins with the HRGPs seem to favor a structural role, although other roles suggested by Mathieu-Rivet *et al.* (2020) cannot be ruled out. Our results unambiguously show strong interactions between mannose rich-molecules and proteins classically described as playing a structural role (See **Figure 5c-d** for the final model involving both HRGPs and LWGPs). We believe that this result provides novel data and new perspective on the complex macromolecular organization of *C. reinhardtii*'s cell wall and of similar microalgae. Whether these mannose-rich proteins act as structural binders between HMW proteins, as we suggest in the manuscript, or play a different role, does not change the fact (which we evidence in this work) that they strongly interact with cell-wall constituents.

Considering all these points, we have purposefully avoided multiple purification steps in order to keep the cell-wall extract as native as possible while still extensively rinsing our extracts, and

we believe those additional glycoproteins are therefore naturally part of, bound to, or at least strongly interacting with our extracted cell wall.

We have added the following paragraph on **page 8**:

*“The Man depletion after cell wall extraction and in the *bald2* mutant (**Supplementary Tables 6-7**), together with the MAS-DNP detection of Man signals in the intact cell walls, are strong indications that Man-rich glycoproteins are part of the cell wall, or at least strongly associated with it.”*

And **page 11**:

“LWGPs as well as “14-3-3” proteins, could be considered as “non-canonical cell-wall proteins”^{41,42}, but they do survive the extraction protocol, two dialysis steps, renaturing non-chaotropic conditions and centrifugation. They should therefore be considered either as part of the cell wall, or strongly interacting with it”

2) Another concern is the lack of statistical evaluation of the quantitative data in this manuscript.

We would like to thank the Reviewer for raising this point, which gives us the opportunity to clearly indicate the number of biological replicates we have made. We have now added a table with the full list of strains and replicates studied (**Table S21**, Supplementary information). Cell-wall extraction followed by SDS-PAGE and ssNMR analysis was performed on eleven fully independent samples (new cell culture batch ran on different days using different pre-cultures) from *wt* and *bald2* strains, and gave almost identical results.

Since ssNMR complete and precise quantification is very time consuming, it has been performed on one sample only. Nevertheless, all 1D and 2D spectra were overlaid and showed the same intensities for each peak or cross peak. Furthermore, ssNMR consists in detecting signals from milligrams of samples, therefore NMR detects a statistical average of billions of molecules.

GCMS has been used to assess glycan composition in different samples, including extracts and whole cells. For example, comparing the *bald2* and *wt* cell wall extracts, the glycan composition was found to be very similar (~1% variability between *wt* and *bald2* strains), and compatible with the NMR results.

Finally, HPLC has been performed on different cell-wall sample replicates (two *wt* and two *bald2* cell-wall extracts), giving again high reproducibility, with a standard error inferior to 0.5%, as reported in **Supplementary Table S11**.

As the Reviewer and future readers can now assess, our data are reproducible and statistically relevant.

3) Based on **Fig. 3c** showing that Threonines interact mainly with Mannose, should we consider whether Man-O-Thr is present in the *Chlamydomonas* cell wall?

I suggest the authors explain the evidence supporting the identification of O-glycosylated threonine in the main text using one of the main figures or tables.

The authors must thoroughly consider other information on whether to support the O-glycosylated threonine and which sugar is linked to threonines in the Chlamydomonas cell wall.

The Reviewer raises an interesting point since our study of these newly found or poorly studied LWGPs raises new questions on glycan and amino acid composition, as well as glycosylation schemes.

It should first be clarified that our NMR results show spatial proximity between Threonine and Mannose, but not a direct proof of glycosylation. Threonine and Mannose could either be spatially or covalently linked. We did not have a full proof of the O-glycosylation of Threonine, although our data seemed to support this hypothesis. It is thus important to consider other information such as the one available from the genome and the few analytical studies that have attempted to characterize glycosylation in *C. reinhardtii* or close organisms.

Glycosylation of Threonine in microalgae is not unusual as it has been observed by Balshüsemann and Jaenicke (1990) in the Chlorophyte *Volvox carteri f. nagariensis*, a green algae closely related to *C. reinhardtii* – also a Chlorophyta.

In *C. reinhardtii*, proteins involved in transducing mechanical stress of the cell wall have been identified, and they are analogous to proteins with Ser/Thr-rich extracellular domain in yeast decorated with O-mannosylations (Cronmiller *et al.* 2019).

We carefully searched the *C. reinhardtii* genome (<https://phycocosm.jgi.doe.gov/>) and indeed, many galactosyl transferases have been identified whose substrate are unknown, many of them having Mannose as potential donors, and some of them having Serine or Threonine as potential acceptors. Interestingly, two *C. reinhardtii* galactosyl transferase families are absent in *Arabidopsis thaliana*, making them good candidates for a role in *C. reinhardtii* cell wall formation. One of them (GT105) has a domain of unknown function DUF1736, which is “found in O-mannosyl-transferases TMTC1-4” and TMTCs transfer mannosyl residues to the hydroxyl group of Serine or Threonine (see www.ebi.ac.uk/interpro/entry/pfam/PF08409/ or www.uniprot.org/uniprotkb/A0A2K3E6S7/). This domain is found in many organisms such as bacteria, diatoms, algae and even humans (Larsen *et al.* 2019, Eisenhaber *et al.* 2021). It is well aligned in *C. reinhardtii* and corresponds to a single gene in the alga genome (https://phytozome-next.jgi.doe.gov/report/transcript/Creinhardtii_v5_6/Cre01.g031800.t1.1). This GT105 is therefore a potential candidate to explain the presence of Man-O-Thr in our LWGPs, although this is just a hypothesis. This shows how little is known on glycosylation in *C. reinhardtii*, and new discoveries will help better understand this key process in cell-wall formation in the future.

To answer the Reviewer’s question, we have sent our sample to Professors Adnan Halim and Hiren Joshi at the University of Copenhagen, who are specialized in the study of protein O-mannosylation using lectin capture and advanced mass spectrometry. Not only have they found mannosylated proteins in our cell wall extract, they have also identified mannosylation sites on both Serine and Threonine residues (see the new relevant part in the article, pages 14-15). This information has thus enabled us to update the final model shown in **Figure 5c-d**.

*“The high abundance of Man in C. reinhardtii’s cell wall suggests the presence of O-glycosylation to Thr. To verify this hypothesis, a cell-wall protein extract was digested with trypsin and enriched by Concanavalin A lectin weak affinity chromatography (ConA-LWAC)^{49,50}. Our glycoproteomic analysis identified 18 unique glycopeptide sequences from 11 proteins (**Supplementary Table 14**) using higher-energy collisional activation (HCD) and electron-transfer dissociation (ETciD) MS. The results suggest an O-linked glycosylation of Ser and Thr with hexose modifications, likely α -linked Man or Man₂, thus demonstrating the presence of biosynthetic machineries in Chlamydomonas capable of catalyzing these modifications.*

*The presence of O-glycosylation in C. reinhardtii’s cell wall architecture, notably the O-glycosylation of Thr by Ara in HRGPs previously demonstrated, is confirmed by our MAS-DNP results^{26,51,52}. In addition, our GC-MS analysis of partially methylated alditol acetates (**Fig. 2b**) shows that Man linkages (1,2 and 1,3) are consistent with Man oligosaccharides attached to proteins via N-linked glycosylation²⁷. Since Asn residues are abundant in LWGPs (ca. 15%), and can only be N-glycosylated - a post-translational modification never reported in C. reinhardtii’s cell wall - at least some of the LWGPs GP4 to GP7 should be N-glycosylated via their Asn residues, with GlcNAc linked to Man-rich oligosaccharides⁵³.*

Our results suggest a model in which glycans are spatially isolated from the amino acid residues, and we postulate the presence of abundant but short Man-rich oligosaccharides. We confirmed the O-glycosylation of Hyp (and possibly Ser) to Ara in HRGP, and propose two new glycosylation bonds in LWGPs: an N-glycosylation of Asn with GlcNAc and oligomannoside, and an O-glycosylation of Thr to Man. Finally, we also report new intermolecular contacts between Hyp and Man that would constitute the interface between HRGPs and LWGPs.”

While these results pave the way to a further in-depth characterization of these glycoproteins, a full identification of mannosylation schemes in *C. reinhardtii* glycoproteins is possible but way beyond the scope of this work, as it would involve purifying the cell wall glycoproteins, deglycosylating them, sequencing them, and then using MS to identify the glycosylation patterns. We see this a project in itself that cannot realistically be carried out within this manuscript, but we have added this as a perspective.

I want to see the authors explain that the observed O-glycosylated threonines are unlikely from a contaminated source material included in the final cell wall extracts.

As described in the answer to the first point, we have good reasons to believe that the glycoproteins and glycosylated amino acids that we detect are constitutive of the cell wall or strongly interacting with it. All the points explaining why we consider that these glycoproteins do not originate from contaminated source material apply to the potentially O-glycosylated proteins.

4) The second one is the lack of O-glycosylated serine. There is no serine signal indicated in Figure 3c. However, glucose or rhamnose and mannose/galactose show connectivities to unknown amino acids (first paragraph on page 13). So, the unknown amino acids may include serine. If the

authors disagree, please explain why serine is an unlikely candidate for the unidentified amino acids close to galactose.

We thank the Reviewer for raising this issue. We wrote “unknown amino acid” because we observe an NMR spatial connectivity between a glycan and an amino acid signal, but overlapping and low resolution (**Figure 4c**) prevent us to identify the actual amino acid. It could, indeed, include Serine, which are notoriously difficult to detect by NMR! We have changed **Figure 4c** and added the following sentence on **page 14** :

“Although the glycosylation of Ser is expected¹³, Ser-glycan contacts could not be resolved due to the overlap of Ser C α resonances with those of Hyp and Thr, and that of Ser C β resonances with glycans’ C6.”

Curiously, in the last paragraph of **page 13**, Hyp, Thr, and Asn backbone nitrogens were commented as connected amino acids to glycans. However, **Supplemental figure 8** shows that Hyp/Thr/Ser signals are clustered as if they are indistinguishable. If Serine signals are not found in any dipolar coupling (C-C) or 15N-13C connectivity analysis, please explain how the authors concluded that serine is devoid of glycosylation.

It is critical to carefully address the absence of O-glycosylated serine in the data. Because this is contradictory to Gal-O-serines characterized in the plant cell wall and the serine-specific galactosyl transferase identified from the *Chlamydomonas* cell wall extract (Saito et al., Identification of Novel Peptidyl Serine-Galactosyltransferase Gene Family in Plants. JBC 2014).

Again, Reviewer 1 is right. We could not isolate any unique resolved glycosylated Serine signal, neither by ssNMR nor by MAS-DNP, but our raw data cannot exclude the possibility of glycosylated Serines in the cell wall (**Figure 4c**).

The presence of Serine-galactosyl transferase (SGT) in *C. reinhardtii* (Saito et al. 2014) strongly suggests the presence of glycosylated Serines in the microalga, but not necessarily abundantly in our extracts. In full analogy with Hydroxyprolines, Serines are more abundant in the HRGPs (represent ~around 10% of HRGP’s amino acids, or 2% of total amino acids) than in LWGPs (~4% of LWGP’s amino acids or 3% of total amino acids). Therefore Serine residues appear to contribute less to the overall signal in our complex extracts (5% of total amino acids).

For clarity, we propose a new **Figure 3b** showing the proportion of amino acids in each protein and in the overall cell wall. This convincingly shows that the low amount of Hydroxyprolines and Serine in our cell-wall extract is due to the higher abundance of LWGPs as compared to HRGPs.

We were surprised, however, by the NMR behavior of Serines, which were not as hydrated as other glycosylated amino acids, and no additional Serine signals appeared after deglycosylation. We therefore concluded, probably too hastily, that Serines, since they behaved as unglycosylated amino acids, were indeed not glycosylated.

Following the recommendation of the Reviewer, and also based on the LC-MS results obtained at the University of Copenhagen, we have modified the manuscript (page 16) to confirm the possibility of glycosylated Serines.

“Considering the LC-MS results and the resolution of our MAS-DNP spectra, glycosylation of at least a fraction of Ser residues is highly probable. We explain the fact that Ser and Asn

overall appear less affected by deglycosylation, and that their spatial contacts to glycans are hard to detect, from their dilution by pools of non-glycosylated amino acids.

As mentioned before, the oligosaccharides bound to amino acids in the cell wall extract should include an average of three sugar units. Moreover, we determined a glycan/amino acid ratio of 1:2, which would correspond to 50% glycosylated amino acids with a single glycan or 16% with an average oligosaccharide length of three glycans. Therefore, Hyp and a significant part of Thr, Ser, and Asn residues must be glycosylated. As shown in Fig. 2c and 3a, our results are in good agreement with a model where all Hyp and approximately half of Ser are glycosylated in the form Hyp-O-Ara-Ara-Gal and Ser-O-Ara-Ara-Gal, as suggested in HRGPs²⁶, and potentially Thr-O-Ara-Ara-Gal. In addition, most remaining Thr and Ser residues as well as approximately half of Asn would be glycosylated in the form Thr-O-Man-Man, Ser-O-Man-Man and Asn-N-GlcNAc-Man-Man in the LWGPs. There would be additional minor forms of glycosylated amino acids with Glc, Rha or Xyl glycans, but also possibly shorter, longer or branched oligosaccharides, as long as the ratio between glycans would remain similar^{23,27}."

5) Overall, the manuscript is difficult to comprehend due to the insufficient description of the ssNMR results. Although I do not list here exactly which parts of the figures are not explained well nor mentioned, I want to share the expected difficulty of general readers in the *Nature communications* journal. The difficulty is exacerbated by placing critical information in the supplemental results again without sufficient explanation.

We apologize if the manuscript was hard to read. Our intention was to keep it short while have making our message clear. We have now significantly modified the manuscript, and took particularly great care of the figure captions. They were adapted to the broad readership of *Nature communications*. We hope that the manuscript is now clearer. In addition, we have modified some sections by eliminating NMR technicalities to focus on the results. For example, the section on dynamics (page 18) now reads:

"Rigid glycans and mobile proteins. We then probed the dynamic properties of each chemical site by recording 1D ssNMR spectra with different 1H-to-13C polarization transfer schemes: Cross-Polarization (CP) - which enhances the sensitivity of the rigid molecular segments – and Insensitive nuclei enhancement by polarization transfer (INEPT)^{32,60} for dynamic segments (Supplementary Information). The CP build-up was faster for glycans, indicating a strong dipolar-coupling network, comparable to crystalline starch⁶¹. However, the CP decay was slower for glycans, indicating less motion on the millisecond regime. Glycans with low C1 chemical shifts, i.e. involved in terminal groups, were particularly favored by INEPT. Terminal Man and Gal units are therefore both highly hydrated and very mobile as compared to other polysaccharides, suggesting their presence at the water-amino acid interface.

Motional differences could be more quantitatively determined by measuring ¹H and ¹³C longitudinal relaxation times in the laboratory (T₁) and the rotating frame (T_{1ρ}) (Supplementary Information). No differences were observed in the ¹H T₁ values, indicating

that ^1H polarization is homogenized throughout the sample by spin diffusion. On the other hand, carbohydrate ^{13}C T_1 as well as ^1H and ^{13}C $T_{1\rho}$ values were slightly longer than those of amino acids, showing that glycans are less mobile than proteins, both on the nanosecond and millisecond scales. As an example of these clear discrepancies between glycans and proteins, ^{13}C T_1 s are shown in **Fig. 5b**. The response of carbohydrates and proteins to polarization transfer schemes indicates a higher mobility of glycans, in particular when they are located in terminal groups (see Supplementary Information section for more details).

In conclusion, proteins and glycans differ in both dynamics and hydration, with mobile protein domains, more rigid but hydrated glycan domains, and glycosylated proteins at the interface. These two families of molecules are spatially segregated, even if it is in small nanometer-scale regions that are chemically and dynamically heterogeneous, similarly to layered semi-crystalline polymers or plant cell walls⁶²."

Reviewer 2

1) The authors made an attempt to describe, at atomic-resolution, the cell wall organization of this model system. However, the study falls short in providing new and exciting results compared to the JACS paper published by the same authors in 2021 (*Identification and Quantification of Glycans in Whole Cells: Architecture of Microalgal Polysaccharides Described by Solid-State Nuclear Magnetic Resonance*, DOI: 10.1021/jacs.1c07429).

Although the JACS 2021 paper provided new insights on how solid-state NMR experiments could provide meaningful details to scrutinize algae cell wall, a very similar analytical methodology is used in the present manuscript, again on a model algae system, therefore the study clearly lacks novelty. The composition of the cell wall of this system was already known.

We thank the Reviewer for this comment, which allows us to clarify the focus of the article. This is clearly not a methodological paper, although some of the NMR methods used in this work appear in the JACS paper mentioned above. Here are the main points, which are new in the article: *C. reinhardtii*'s cell wall was, up to now, described as being solely composed of high molecular weight hydroxyproline-rich glycoproteins (HRGPs). In our manuscript, we describe low-molecular weight glycoproteins (that we name LWGPs), which strongly interact with the HRGPs and would be an integral part of the cell wall. The composition of the cell wall is therefore different from what was previously thought. One of our main findings is that LWGPs need to be considered to understand the whole architecture of *C. reinhardtii*'s cell wall - and potentially of the glycoprotein-rich cell wall of other microalgae. Moreover, we further characterize the structure of the HRGPs, quantify their glycosylation (idem for the LWGPs) and provide some information on spatial proximity between glycans and amino acids, their dynamics and hydration. As emphasized by Reviewer 1, this is all new information and even potentially quite controversial.

While this is not a methodology paper, it might be necessary to highlight the differences with the JACS paper. We indeed use the same methodology described in that paper, but here it is applied to glycoproteins with a description of both amino acids and glycans (while the JACS paper only focused on glycans). Protein secondary structure, hydration, dynamics and connectivities were all absent in the previous paper. We also used Dynamic Nuclear Polarization, which was not used in the JACS paper. We would also like to stress that the microalga studied in the JACS paper

(*Parachlorella*) has a cell wall composition based on cellulose, somewhat similar to higher-plant cell walls, whereas the specificity of the *Chlamydomonas* cell wall is that it is composed of glycoproteins with no major large glycan polymer.

We have thus significantly modified the manuscript in response to the Reviewer's concerns, and we hope that the novelty of our findings is clearer. We have also softened any methodological aspects that were not the focus of the manuscript, notably the last sentences of the abstract, which now reads:

"The structural insight exemplifies strategies used by nature to form cell walls devoid of cellulose or other glycan polymers."

and at the beginning and at the end of the discussion:

"This work allowed refining the layered model of C. reinhardtii's cell wall, which was primarily known from electron microscopy. By studying a hydrated cell wall in a near native-environment in comparison with intact cells, we were able to obtain unique structural and dynamic information on the sugar and proteins that compose this complex biological scaffold. These molecular-level data improve the current model of the glycoprotein-rich cell wall of C. reinhardtii, and allows reconciling this organization with the cell wall function of microalgae, and higher plants to a certain extent."

...

"This work is a step towards a better understanding of glycoprotein cell walls in intact cells."

2) It is also unclear how the NMR analysis could help to understand the cell wall life cycle of such algae, e.g. the so-called "cell wall regeneration" would have been an interesting mechanism to explore.

This is a very interesting perspective, which could eventually lead to a research project now that we have more details on the composition and structure of the cell wall. In case this topic might be related to the Reviewer's interests, we might mention that some of our unpublished results on microalgae with different mating types did not reveal striking differences with the cell walls of sexually differentiated cells. This implies that after cell-wall shedding, no major changes can be detected in the composition of the regenerated cell wall central triplet. One experimental difficulty in such a study might be that *Chlamydomonas* cell walls have been reported to regenerate their cell walls in 3 to 4 hours approximately so harvesting enough cell wall might be difficult.

3) The manuscript is well executed from a NMR point of view, and would be more suitable for a NMR-oriented journal or a journal focused on analytical (bio)chemistry.

We would like to thank the Reviewer for the appreciation of the execution of the experiments. In the revised manuscript, we have enriched the non-NMR analysis and elaborated on the background and novel findings regarding glycoproteins, algal strains, and cell wall structure. We hope the revised manuscript can be more accessible to the broad readership in the field of

analytical chemistry, macromolecular structure, plant biology, and biochemistry to fit the wide audience of *Nature Communications*.

4) *The last sentence of the abstract highlights that the proposed methodology could be “transposable to study other microorganisms and plant materials”, however this solid-state NMR methodology has been already used many times for fungal micro-organisms by one of the corresponding authors (T. Wang), not mentioning all the work from his previous supervisor (Mei Hong) on plant cell wall using NMR. This sentence should be changed.*

Thank you. As the focus of the article is not the methodology, we have deleted this sentence following the Reviewer’s advice. We would like to point out that even though previous studies have indeed used this ssNMR toolbox for different cellular samples, we have enriched its application with a focus on the glycoproteins, which is of interest to many potential readers, not only in the field of microalgae and plants.

Reviewer 3

1) *My major concern is that the authors primarily investigated the architecture by reconstituting cell wall extracts. It is not clear how representative this is of the native cell wall architecture/organization. Despite the 1D and 2D PDSO ssNMR (Fig. 1e & 1f) showing similar peaks between whole cell and cell wall extracts, the extracted cell wall components may not reorganize as if they were in the native form, and this may not be reflected by the (Fig. 1e & 1f) comparison. In particular, the PDSO experiments were later used for glycan-protein contact analysis and peak intensities were used for quantification. Authors need to better clarify in the discussion how they ensure that the extracted cell wall architecture represents their native form. Maybe inclusion of a comparison between the difference spectra of “whole cell” and “whole cell cw-extracted” vs the “cell wall extract” spectra can help.*

We thank Reviewer 3 for the careful reading, insightful comments and for this question that we have taken into account from the beginning of this research. We have partially answered this question in our answer to Reviewer 1 by showing that LWGPs are part of the cell-wall environment in the cell. Sodium perchlorate being a chaotropic agent, it could indeed denature insoluble proteins before dialysis, but the protocol we use has shown that dialysis was sufficient to reconstitute the cell-wall architecture, as shown by deep-etch SEM (Goodenough *et al.* 1985). We have also expanded the description of the cell wall by providing a dedicated subsection “*Cell wall extraction retains the macromolecular core*”.

We would also like to thank the Reviewer for kindly suggesting to compare the whole cell with a sum of the cell-wall extracted cell and the cell-wall extract. We have therefore made a new figure (Figure 1e), showing a very good match between the whole cell spectrum and the reconstituted cell spectrum (made by combining the cell-wall extracted cell spectrum and the cell-wall extract spectrum, with a 62/38 intensity ratio). These 1D ssNMR spectra are the most reliable in terms of quantification. The only change that we can detect is in the lipid content before and after the cell-wall extraction, which is explainable by the metabolic stress induced by

this kind of treatment, even if the cell surface membrane remains intact as demonstrated by electron microscopy.

2) Another suggestion to the authors is that they need to provide more background information on a few topics/concepts that are discussed in the results section, which can greatly help the audience follow the story. Technical terms were not introduced but brought up in later results section, which make the story hard to follow. For example, introduction to the current understanding of the composition, structure and organization of *C. reinhardtii* glycoprotein-rich cell wall. More introduction/definition of the HRGPs, and low molecular weight glycoprotein (14-3-3) is needed, including brief descriptions of core structures, and what is the current understanding of their functional roles in the algal/plant cell wall. Introduction of N- and O-linked glycosylation would also be beneficial.

We thank Reviewer 3 for this suggestion and have better described the current knowledge on HRGPs and their O-glycosylation, and introduced its difference with N-glycosylation. The paragraph on **page 3** now reads:

*“Previous investigations of *C. reinhardtii* cell wall provided a partial sequencing and composition of HRGPs by chromatography and mass spectrometry (MS), while EM revealed the layered architecture. HRGPs were shown to be glycosylated by short O-oligosaccharides containing 1-5 residues in which arabinose (Ara) and galactose (Gal) are the most abundant. These oligosaccharides would be linked to HRGP domains containing Ser or polyproline helices^{19,23-25}. Traces of mannose (Man) were also reported, accounting for up to 7% of the overall glycan content, as well as β (1-2)-linked L-Arabinose (Ara) disaccharides substituted with galactofuranoses (GalF) and O-methylation²⁶. While O-glycosylation is the only post-translational modification described so far in the cell wall, an N-glycosylation pathway has also been described in *C. reinhardtii* that involved the linking of two N-acetylglucosamines (GlcNAc) and two Man to an asparagine (Asn)²⁷.”*

We have also specified that the 14-3-3 proteins were not as well characterized (also on **page 3**):

“These “14-3-3” proteins are rich in serine (Ser) and threonine (Thr), but not fully sequenced²².”

In **Fig. 2b** caption, authors mentioned the *bald2* strain was used for the ssNMR study and more details were not provided until the method section. This needs to be brought up to early text in the results section or the introduction. And description of this strain and why using this strain for the study needs to be discussed.

We thank the Reviewer for this comment, which allowed us to add this information earlier in the text. On **page 6**, in the results section, we now specify why we use the *bald2* strain:

*“**Glycan composition of the cell-wall extract.** We determined the complete profile of glycans and amino acids in *C. reinhardtii*'s cell wall extracts, using the flagella-deficient *bald2* strain, to avoid the large flagella-associated proteins (~200 kDa) that have been shown to contaminate cell-wall extracts during the growth cycle.”*

3) Some statements need references to support them e.g “that of *C. reinhardtii* is built on glycoprotein-rich layers that can be compared to extensins in higher plants”.

We have included a new reference to corroborate this statement made by Ferris *et al.* (2001), but Saito *et al.* (2014) also considers this comparison legitimate.

4) **Fig. 1e** is hard to understand. In the text (*line 86* and the **SI Fig.1** caption), authors described MAS-DNP is required cryogenic temperature, but why collected the spectra separately for MAS-DNP and cryogenic temperature? It seems the figure caption is wrong.

Blue spectra are MAS-DNP with cryogenic temp., and the orange spectra are collected without MAS-DNP. Authors need to clarify this, and clearly label the spectra.

Figure 1e has been modified to address this comment. To answer the Reviewer’s comment, we just used the low temperature hardware of the MAS-DNP without using any microwave signal-enhancement. This allowed us to evaluate the effect of low temperature on spectral resolution and sensitivity, as seen in the updated **Supplementary Figure S1**.

5) *Line 94 and Fig. 1g*, authors need to provide more description on the cell-wall deficient (*cw15*) strain, and explain in the text how it should be compared to the “whole cell cw- extracted” spectrum.

This has been clarified in the corrected manuscript by modifying the following paragraph on **page 5**:

“The cw15 mutant has compromised cell wall synthesis⁸. The remaining wall components resemble the cell wall outer layer³⁰, resist chaotropic extraction³¹ and therefore likely correspond to the residual chaotrope-insoluble W1, W2 and W7 layers. The resemblance between the MAS-DNP spectra of cw15 cells and the wt sample after cell-wall extraction further confirms that our extracts are composed of the cell wall central triplet.”

6) *Line 115-118*, authors didn’t specify how the SDS-PAGE indicate similar glycosylation level (**Fig. 2b**) as results indicated by ssNMR (**Fig.2a**). The SDS-PAGE results discussed in the earlier two sentences don’t provide direct link to conclude the last sentence of the paragraph. As the authors state in *Line 218*, the caption of **Fig. 2b** states “lane 6 : deglycosylated 13C-labeled bald2 cell wall with tentative molecular weight assignment;” it seems unclear how much glycan was removed. The Authors need to clarify why they conclude that the SDS-PAGE indicate similar glycosylation level (**Fig. 2b**) as results indicated by ssNMR (**Fig.2a**), or these areas in the text need to be revised

We would like to thank the Reviewer for this comment. Indeed, the qualitative assumption that glycosylation levels are similar are based on band assignment done considering their relative intensities. We expect them to be the same before and after deglycosylation since Coomassie

blue does not stain glycans decorating the proteins. Since the glycosylation levels were just qualitatively estimated, we have moderated the corresponding sentence in the manuscript.

To better identify the protein bands, amino acid analysis of each band after deglycosylation would have been interesting, but it was outside the scope of this work. As requested, we have also added the Supplementary **Figure S16** showing the uncut SDS-page gels. The paragraph has been changed to:

“Interestingly, the deglycosylation led to a shift of all SDS-PAGE protein bands, proving that the low molecular weight proteins are also glycosylated (Fig. 2a, lane 5 vs. 6) at an approximate level of 20 % assessed by the change in molecular weight.”

7) In the sentence from Line 119-124 and the caption of **Fig 2c**, authors need to specify that the “extracted cell wall” is analyzed.

This has been clarified on **page 6** (“reconstituted cell wall”) and in the figure caption (“extracted” cell wall) of the corrected manuscript.

8) Line 131, typo in “groups of oligosaccharides (blue bars on **Fig. 2e**).”. There are no blue bars in **Fig. 2e**

This has been changed in the corrected manuscript.

9) Even though the authors specified the low percentage of HRGPs (20%) of total protein content, considering a third of HRGPs should be Hyp, as the authors also stated in the text, detected Hyp is surprisingly low (~1%). Can the authors provide more explanation on the low abundance of Hyp detected by HPLC and ssNMR?

There are two different numbers. One refers to the amount of Hyp detected by NMR, which is very low, both because of its relatively low abundance and its dynamics. Then there is the amount of Hyp as measured by HPLC, which is indeed low, but consistent with published data. Hyp represents:

- 20% of the largest HRGP (GP1), which amounts to 5% of the proteins in our cell-wall extract;
- 15% of the second HRGP (GP2), which represents 7% of the cell-wall proteins;
- 6% of the third HRGP (GP3), which represents 9% of cell-wall proteins);
- Less than 1% of the LWGPs, which overall represent almost 80% of proteins.

In total, this $[(0.05 \times 0.2) + (0.07 \times 0.15) + (0.09 \times 0.06) + (0.8 \times 0.01)]$ predicts an expected 3% of Hyp, which coincides with the HPLC measurement, within experimental biological error.

We agree this was not clear and have thus clarified the manuscript. In particular, we have added pie charts describing the relative abundance of proteins and amino acids in each protein in **Figure 3b**.

10) **Fig. 3a** and **SI Fig. 6**, spectra legend should be corrected to “extracted cell wall” vs “deglycosylated extracted cell wall”. Also, for **Fig. 3a-c**, spectra axis labeling is covering the tick marks.

It is now updated in the current version for both the main text and the Supplementary document.

11) Sentence in line 248-250, “Glycans thus appear to have ... proteinated content of the cell wall” is hard to parse, and would benefit from rephrasing.

We have rewritten the sentence as:

” Altogether, glycans appear to have almost no influence on the overall protein structure and dynamics, suggesting a model in which they are spatially isolated from the amino acids, in sugar domains outside or within protein segments.”

12) Line 295, “allow determining” to “allow to determine”

Done.

13) Line 299, “**Fig. 3d** and **Supplementary Fig. 7**” should be “**Fig. 3d** and **Supplementary Fig. 8**”

Done.

14) Line 309, “long oligosaccharides” should be changed to “long polysaccharides” to be consistent with later text.

This whole section has been modified.

15) Line 350-352, sentence, “Moreover, the residues that can be glycosylated, namely Hyp, Thr and, to a lesser extent, Asn and Ser, are hydrated to much higher levels close to those of glycans’.”, seems reversed.

This sentence is correct, as can be seen from **Figure 5a**: Hyp and Thr are more hydrated than “regular” amino acids, and so are Asn and Ser, to a lesser extent. Since this sentence was not clear, we have rephrased it as:

”Moreover, the residues that can be glycosylated, i.e. Hyp, Thr and, to a lesser extent, Asn and Ser, have a much higher level than other amino acids, which almost reaches that of glycans.”

16) Line 397, "...detected by CP-PDSD, or at low temperatures (< -160°C) (**Supplementary Fig. 2**)" seems wrong figure indicated. Authors meant the **Fig.3b&c** in the main text?

Reviewer 3 is right. Correction done.

17) Line 452, typo "macroalgae"

Done.

References

- Balshüsemann, D. and L. Jaenicke (1990). "The oligosaccharides of the glycoprotein pheromone of *Volvox carteri* f. *nagariensis* lyengar (Chlorophyceae)." *Eur. J. Biochem.* **192**(1): 231–237.
- Cronmiller, E., D. Toor, N. C. Shao, T. Kariyawasam, M. H. Wang and J.-H. Lee (2019). "Cell wall integrity signaling regulates cell wall-related gene expression in *Chlamydomonas reinhardtii*." *Sci. Rep.* **9**(1): 12204.
- Eisenhaber, B., S. Sinha, C. K. Jadalanki, V. A. Shitov, Q. W. Tan, F. L. Sirota and F. Eisenhaber (2021). "Conserved sequence motifs in human TMTC1, TMTC2, TMTC3, and TMTC4, new O-mannosyltransferases from the GT-C/PMT clan, are rationalized as ligand binding sites." *Biol. Direct* **16**(1): 4.
- Ferris, P. J., J. P. Woessner, S. Waffenschmidt, S. Kilz, J. Drees and U. W. Goodenough (2001). "Glycosylated polyproline II rods with kinks as a structural motif in plant hydroxyproline-rich glycoproteins." *Biochemistry* **40**(9): 2978–2987.
- Goodenough, U. W., B. Gebhart, R. P. Mecham and J. E. Heuser (1986). "Crystals of the *Chlamydomonas reinhardtii* cell wall: polymerization, depolymerization, and purification of glycoprotein monomers." *J. Cell Biol.* **103**(2): 405–417.
- Goodenough, U. W. and J. E. Heuser (1985). "The *Chlamydomonas* cell wall and its constituent glycoproteins analyzed by the quick-freeze, deep-etch technique." *J. Cell Biol.* **101**(4): 1550–1568.
- Goodenough, U. W. and J. E. Heuser (1988). "Molecular organization of the cell wall and cell-wall crystals from *Chlamydomonas eugamatos*." *J. Cell Sci.* **90**(4): 735–750.
- Hills, G. J., J. M. Phillips, M. R. Gay and K. Roberts (1975). "Self-assembly of a plant cell wall *in vitro*." *J. Mol. Biol.* **96**(3): 431–441.
- Jamet, E., C. Albenne, G. Boudart, M. Irshad, H. Canut and R. Pont-Lezica (2008). "Recent advances in plant cell wall proteomics." *PROTEOMICS* **8**(4): 893–908.
- Jiang, K.-s. and G. A. Barber (1975). "Polysaccharide from cell walls of *Chlamydomonas reinhardtii*." *Phytochemistry* **14**(11): 2459–2461.
- Larsen, I. S. B., Y. Narimatsu, H. Clausen, H. J. Joshi and A. Halim (2019). "Multiple distinct O-Mannosylation pathways in eukaryotes." *Curr. Opin. Struct. Biol.* **56**: 171–178.
- Mamedov, T. and V. Yusibov (2011). "Green algae *Chlamydomonas reinhardtii* possess endogenous sialylated N-glycans." *FEBS Open Bio.* **1**: 15–22.
- Mathieu-Rivet, E., N. Mati-Baouche, M.-L. Walet-Balieu, P. Lerouge and M. Bardor (2020). "N- and O-glycosylation pathways in the microalgae polyphyletic group." *Front. Plant Sci.* **11**.
- Saito, F., A. Suyama, T. Oka, O. T. Yoko, K. Matsuoka, Y. Jigami and Y. I. Shimma (2014). "Identification of novel peptidyl serine α -galactosyltransferase gene family in plants." *J. Biol. Chem.* **289**(30): 20405–20420.
- San Clemente, H., H. Kolkas, H. Canut and E. Jamet (2022). "Plant cell wall proteomes: the core of conserved protein families and the case of non-canonical proteins." *Int. J. Mol. Sci.* **23**(8): 4273.
- Vogeler, H.-P., J. Voigt and W. König (1990). "Polypeptide pattern of the insoluble wall component of *Chlamydomonas reinhardtii* and its variation during the vegetative cell cycle." *Plant Sci.* **71**: 119–128.

Voigt, J., I. Liebich, M. Kieß and R. Frank (2001). "Subcellular distribution of 14-3-3 proteins in the unicellular green alga *Chlamydomonas reinhardtii*." Eur. J. Biochem. **268**(24): 6449–6457.

REVIEWER COMMENTS

Reviewer #1 (Remarks to the Author):

I found the revised manuscript addressed all of my concerns. So, the revisions are satisfactory. In particular, the newly added mass spectrometry analysis of mannosylated proteins provided evidence for the existence of O-mannosylated threonine/serine in *Chlamydomonas* cell walls.

Central to the author's argument is the contribution of mannose to the strong interaction between the two pools of cell wall proteins, HRGPs and non-HYP glycoproteins as LWPs in this manuscript.

The contribution of Mannose to the *Chlamydomonas* wall is evidenced by NMR data and also by the high quantity of mannose in the authors' cell wall preparation (Fig. 2d) and the 10% of GP3 glycoprotein mass as mannose reported in Kilz et al. (2000, referenced in the manuscript). Given the predictably significant contribution of mannose in cell wall preps, I would expect to see major wall proteins, GP3 from HRGP pool and GP4 ~ GP7 in LWP pool to be detected in the new MS analysis.

So, I double-checked the identified peptide data in Supp. table 14. The doublecheck leaves critical questions that I want to ask the authors for clarification.

1. The identified UniProt entries according to the mass profile are all from *Chlamydomonas incerta* proteins not from *C. reinhardtii* ones. Is there a chance that *C. incerta* cells were harvested for the mass spectrometry analysis? If not, an additional explanation in the method section will be advised. ^[1]_[SEP]
2. The *C. reinhardtii* homologs to the identified *C. incerta* proteins are mostly cell surface proteins potentially involved in signaling (having predicted transmembrane domains at the C terminus), with or without proline-rich segments. So, the question is whether the analysis method adopted here is inherently biased toward non-structural cell wall proteins. Or, it might be the case that most of the mass profiles could not be further analyzed for their corresponding peptides due to complex glycosylation patterns to be modeled. ^[1]_[SEP] If the listed peptides correspond to the major fragments (should be detected multiple times), I want to ask if some of the identified peptides should represent GP4 ~ GP7. ^[1]_[SEP]
3. Given the disparity between the major contribution of mannose in chemical analysis and the marginal detection of mannosylated peptide in mass spectrometry analysis, I am cautious about the significance of mannose-rich glycoprotein in the *Chlamydomonas* wall architecture.
4. So, the significance of mannose contribution toward the *Chlamydomonas* cell wall architecture may be validated by another approach or should be toned down. For example, authors may examine if mannose-specific antibodies can stain the *Chlamydomonas* wall. Maybe, the mannose-cleaving enzyme treatment

would collapse the Chlamydomonas wall architecture at the EM-level and eliminate or weaken the mannose signal in NMR. (A relevant approach may be found in <https://doi.org/10.1111/j.1365-313X.2010.04319.x>).

Reviewer #2 (Remarks to the Author):

The authors have introduced new data to their manuscript. I'm however still not convinced by the true novelty of the manuscript with respect to their previous JACS paper on the same topic. I still recommend a submission to a more NMR-oriented journal.

If the manuscript is accepted at Nat. Comm, 3 points should be addressed before publication:

Major: their response to the major criticism of the reviewer #3 is not satisfactory. The comparison of 2D ¹³C-¹³C correlation experiments (figure 1) recorded under cryogenic conditions (DNP) is not 100% fair, because the spectral resolution is highly compromised in this condition. Since the authors have recorded 2D ¹³C-¹³C DARR at room temperature, why not showing them in Fig1 ? It will provide a more informative spectral comparison.

Minor: The authors have re-organized the text and give more explanation about the interpretation of NMR results, nevertheless interesting discussion are still included in the SI. E.g: in page 10, 14 and 15 of the SI. I find quite annoying to not have this discussion (although quite technical) in the main text, maybe it can be introduced in the main text in a reduced form. Otherwise the manuscript, in the current form, is still difficult to digest for non-experts of solid-state NMR spectroscopy.

Minor: please split the figure S3 into 3 SI figures, to provide larger figures to the readers. In the current form, this figure and the assignments are impossible to read.

Reviewer #4 (Remarks to the Author):

All my comments in the first review were fully addressed by the authors, I do not have any additional comments

Response to the Reviewers

We would like to thank the Reviewers for the thorough revision of our manuscript. Their comments enabled us to improve our manuscript. All comments have been considered and the corresponding revisions have been highlighted in green within the document.

Reviewer #1 (Remarks to the Author)

I found the revised manuscript addressed all of my concerns. So, the revisions are satisfactory. In particular, the newly added mass spectrometry analysis of mannosylated proteins provided evidence for the existence of O-mannosylated threonine/serine in *Chlamydomonas* cell walls.

Central to the author's argument is the contribution of mannose to the strong interaction between the two pools of cell wall proteins, HRGPs and non-HYP glycoproteins as LWPs in this manuscript.

The contribution of Mannose to the *Chlamydomonas* wall is evidenced by NMR data and also by the high quantity of mannose in the authors' cell wall preparation (**Fig. 2d**) and the 10% of GP3 glycoprotein mass as mannose reported in Kilz *et al.* (2000, referenced in the manuscript). Given the predictably significant contribution of mannose in cell wall preps, I would expect to see major wall proteins, GP3 from HRGP pool and GP4 ~ GP7 in LWP pool to be detected in the new MS analysis.

So, I double-checked the identified peptide data in **Supp. table 14**. The doublecheck leaves critical questions that I want to ask the authors for clarification.

1. The identified UniProt entries according to the mass profile are all from *Chlamydomonas incerta* proteins not from *C. reinhardtii* ones. Is there a chance that *C. incerta* cells were harvested for the mass spectrometry analysis? If not, an additional explanation in the method section will be advised.

Reviewer#1 is right; our database search included all *Chlamydomonas* species. We have now narrowed down our search and modified **Supplementary Table 14** to include only *Chlamydomonas reinhardtii*'s proteins, which left 9 unique peptide sequences from 6 proteins.

2. The *C. reinhardtii* homologs to the identified *C. incerta* proteins are mostly cell surface proteins potentially involved in signaling (having predicted transmembrane domains at the C terminus), with or without proline-rich segments. So, the question is whether the analysis method adopted here is inherently biased toward non-structural cell wall proteins. Or, it might be the case that most of the mass profiles could not be further analyzed for their corresponding peptides due to complex glycosylation patterns to be modeled.

If the listed peptides correspond to the major fragments (should be detected multiple times), I want to ask if some of the identified peptides should represent GP4 ~ GP7.

None of the identified protein in UniProt is annotated as a cell-wall protein, showing the limitation of the current MS approach when dealing with N-glycan/Man-rich complex glycoproteins. As noted by Reviewer#1, cell wall proteins are very difficult to detect by MS for several reasons:

- 1) As for any proteomics approach, abundant proteins are more readily identified.
- 2) The digestion process favors small soluble proteins, and does not favor large non-structured cell-wall proteins.
- 3) The digestion process favors regions either dense with Lys/Arg, or devoid of Lys/Arg. In the case of HRGPs, which are large proteins poor in lysine residues, trypsin digestion will be less efficient.
- 4) As noted by the Reviewer, complex glycosylation patterns will make mass profiles difficult to analyze. This will be the case for mannose residues that could be bound to other sugars, methylated or bear other types of modifications.
- 5) N-glycosylation is not accounted for by this method.

3. Given the disparity between the major contribution of mannose in chemical analysis and the marginal detection of mannosylated peptide in mass spectrometry analysis, I am cautious about the significance of mannose-rich glycoprotein in the *Chlamydomonas* wall architecture.

The Reviewer raises a fair point, which led us to tone down this aspect in our manuscript. We had tried to detect mannosylated cell-wall proteins by MS, but this approach failed for the reasons mentioned above. While we cannot definitively prove that cell-wall proteins are mannosylated, we cannot dismiss this possibility either. The MS methods utilized confirmed the existence of the mannosylation machinery in *C. reinhardtii*, which could potentially be used for cell-wall proteins. Furthermore, the proteins identified in our work are challenging to study using conventional methods. However, ss-NMR has the capability to detect and quantify amino acids and glycans with a relatively simple sample preparation, providing valuable information that would otherwise be unattainable. Considering previous studies that have proved that mannosylation was found in *C. reinhardtii* and N-glycosylation machinery in its cell wall (Mathieu-Rivet *et al.* 2013, Mathieu-Rivet *et al.* 2020), along with our ssNMR and DNP analyses, controls (involving different strains and cell differentiation), and the correlation between extracts and whole cells, we believe that our findings introduce a new proposition that deserves to be presented.

4. So, the significance of mannose contribution toward the *Chlamydomonas* cell wall architecture may be validated by another approach or should be toned down. For example, authors may examine if mannose-specific antibodies can stain the *Chlamydomonas* wall. Maybe, the mannose-cleaving enzyme treatment would collapse the *Chlamydomonas* wall architecture at the EM-level and eliminate or weaken the mannose signal in NMR. (A relevant approach may be found in <https://doi.org/10.1111/j.1365-313X.2010.04319.x>).

We thank the Reviewer for suggesting those experiments that may help proving the presence of mannosylated cell-wall proteins in *C. reinhardtii*. Unfortunately, the only reported antibody that recognizes α -mannose linked to threonine is not commercially available. Alternatively, lectin blots may be used, but lectins would not be able to differentiate a mannose linked to threonine to one found as a terminal structure. Jack bean α -mannosidase could be used to cleave mannose sugars from proteins, but this assumes that there are no "unknown" modifications on the mannoses that may block the enzyme, which is specific for α -mannose itself.

We also agree with the Reviewer that further biochemical assays could confirm the role of mannose in the cell wall architecture. The prevalence of mannanases in plants suggests that microalgae may likewise possess mannose-rich components. However, we are uncertain whether these carbohydrates are pure mannane or mannose-rich polysaccharides. Therefore, such an assay could potentially generate an additional manuscript focusing on understanding, possibly through ssNMR, the effects of mannose removal from the glycoproteins with this type of treatment.

We consequently moderated the conclusions of our manuscript by employing cautious language. Nonetheless, we retained the final *C. reinhardtii* cell wall model, as intermolecular contacts between mannose and hydroxyprolines were unambiguously detected using various approaches, including MAS-DNP.

The presence of mannose-rich low molecular weight proteins in the cell wall is now described as being “likely” in the abstract, and the following paragraph has been added in page 15:

“Although none of the identified proteins is annotated as a cell wall protein, it should be noted that the approach described above favors abundant, small, soluble proteins with simple glycosylation patterns. Large cell wall proteins with complex glycosylation patterns are notoriously difficult to detect by this approach without using specific molecular separation techniques, enrichment and targeted detection and analysis (Banazadeh et al. 2017, Tabang et al. 2021), but are amenable to ssNMR methods as those used in this study. To further ascertain the high mannosylation of C. reinhardtii’s cell wall, our ssNMR approach could be applied on cell-wall extracts from which mannose residues would be removed by mannosidase and other enzymes (Morales-Quintana et al. 2023). Fluorescence imaging using mannose-specific antibodies followed by enzymatic treatment (Marcus et al. 2010) could also be considered.”

Reviewer #2 (Remarks to the Author)

The authors have introduced new data to their manuscript. I'm however still not convinced by the true novelty of the manuscript with respect to their previous JACS paper on the same topic. I still recommend a submission to a more NMR-oriented journal.

The focus of our manuscript is to provide unprecedented atomic-level details on the architecture of a glycoprotein-rich microalgal cell wall. It nicely follows articles studying the cell wall of higher plants published in Nature Communications, and highlights a very different cell wall formation strategy, which does not rely on lignin (or cellulose) but on glycoproteins. The article the Reviewer refers to was primarily methodological, and focused on an engineered microalga, *Parachlorella beijeirinkii*. It did not provide specific information on its cell wall, which was, in that case, cellulose-based and not made of glycoproteins.

Therefore, the targeted readership of our manuscript is broader, and ranges from plant and cell biologists to microalgae-focused engineers. It should also garner the attention of structural biologists interested in glycoprotein-based assemblies, which we consider currently under-researched.

This is probably a minor point, but neither MAS-DNP, nor water-edition methods have ever been applied to algal extracts or whole microalgae. We find this strategy particularly suited to study an

aquatic microorganism, where water is most likely involved in specific domains of the cell wall architecture, as demonstrated in our manuscript.

If the manuscript is accepted at Nat. Comm., 3 points should be addressed before publication:

Major: their response to the major criticism of the reviewer #3 is not satisfactory. The comparison of 2D ^{13}C - ^{13}C correlation experiments (figure 1) recorded under cryogenic conditions (DNP) is not 100% fair, because the spectral resolution is highly compromised in this condition. Since the authors have recorded 2D ^{13}C - ^{13}C DARR at room temperature, why not showing them in Fig1 ? It will provide a more informative spectral comparison.

We thank the Reviewer for the careful reading of our manuscript. The rationale behind the choice made in **Figure 1** is that, despite lower resolution, MAS-DNP show signals from the outer part of the cell **only** that can be more readily compared to the spectra from the cell-wall extracts. As shown before in bacterial systems (Takahashi *et al.* 2013), because the radical used in DNP is unlikely to cross the lipid membrane, DNP can be used not only as signal-enhancement technique but also as a spatially selective strategy.

As suggested by the Reviewer, a comparison between higher-resolution spectra obtained at 283 K can also be attempted. As expected, non-spatially selective whole-cell ^{13}C - ^{13}C ssNMR spectra tend to be crowded, especially with dipolar experiments such as DARR or PDSO. Nevertheless, we can compare the ^{13}C - ^{13}C DP-INADEQUATE ssNMR spectra of the whole cell and cell-wall extract recorded at 283 K. At this temperature, all glycans are detected quantitatively, and we know from GC-MS analysis that *ca.* 40% of them are in the cell wall, but 60% are elsewhere in the cell (**Supplementary Table 7**). We therefore expect at best a maximum 40% overlap of the two spectra. As shown in the **new Supplementary Figure 2**, all the peaks obtained on the extract are present in the whole-cell spectra, and an approximate 40% overlap in the glycan region is obtained.

We consider that the MAS-DNP spectra in Figure 1 more effectively demonstrate the quality of the extraction-reconstitution process. Nevertheless, we have included the spectra requested by the Reviewer in the Supplementary Information (Figure S2) and we refer to them in the manuscript.

Minor: The authors have re-organized the text and give more explanation about the interpretation of NMR results, nevertheless interesting discussion are still included in the SI. E.g: in page 10, 14 and 15 of the SI. I find quite annoying to not have this discussion (although quite technical) in the main text, maybe it can be introduced in the main text in a reduced form. Otherwise the manuscript, in the current form, is still difficult to digest for non-experts of solid-state NMR spectroscopy.

Thank you for this comment. To maintain the article's brevity and ensure readability for a wide audience, we had to keep only the key points of the experiments mentioned by the Reviewer in the manuscript. However, the information presented in the SI is now concisely integrated in the main text. The corresponding experimental conditions are detailed in the Materials and Methods section.

The comparison of the wild-type and flagella-deficient strain is now detailed on **page 6**, and we have removed a sentence on **page 11** to avoid redundancy.

“Our results were independent of the strain type, i.e., with (wt) or without (bald2) flagella, the growth conditions or cell differentiation (sexually competent, differentiated or not), as revealed by the protein composition of different cell wall extracts obtained using polyacrylamide gel

electrophoresis (**Fig. 2a**) or ssNMR (**Supplementary Fig. 3 and Table 12**). The amino acid profile of the protein bands separated by SDS-PAGE was analyzed using HPLC on each cut band. The unlabeled and ^{13}C -labeled cell-wall protein profiles were also comparable using these methods.”

The experiments on **page 16** of the SI (the ^{15}N - ^{13}C correlation experiments) NCaCx are now described on **page 14 of the manuscript**:

“Using an additional ^{13}C - ^{13}C mixing time in an NCaCx experiment, we allowed magnetization to transfer from Ca to surrounding carbons. Connections were thus revealed between O-glycosylated Hyp/Thr’s amides (~115 and ~120 ppm) and several glycans’ C_1 sites, including that of Ara (~109 ppm) (**Supplementary Fig. 13**).”

The experiments on **page 20** of the SI (cross-polarization build-ups) are described on **page 18** and a reference to **Supplementary Figure 16** has been added:

“Despite the low resolution of the 1D spectra, especially in the carbohydrate region, 44 distinct carbon resonances could be identified, i.e., 20 from glycans and 24 from amino acids (**Supplementary Fig. 16 and Supplementary Table 17**).”

“Polarization transfer efficiency differences also explain why amino acids signals, including Hyp cross peaks, were preferentially detected by CP-DARR, or at low temperatures (< -160°C) (**Supplementary Fig. 8-9**). This is most likely due to their unfavorable dynamics or relaxation rates.”

“While T_1 probes motions with correlation times on the order of ns, $T_{1\rho}$ is sensitive to slower (ms) motions. Comparing relaxation times to published values measured in other contexts permits to refine our model.”

The experiments on **page 21** of the SI (^1H T_1 and $T_{1\rho}$ and ^{13}C $T_{1\rho}$) are now summarized on **pages 18-19** of the manuscript and referred to in the manuscript more explicitly:

“Overall, ^{13}C and ^1H relaxation behaviors - both on the ns and ms timescales (**Supplementary Fig. 18**), are not indicative of large crystalline regions but more characteristic of a semi-mobile material akin to surface cellulose or hemicellulose.”

The experiments on **page 19** of the SI (1D experiment and protein/glycan ratio determination) are described on **pages 22-23** and a link to the figure in the SI has been added:

We assess the protein/glycan ratio by using fully relaxed (30 s recycling delay) 1D direct polarization (DP) ^{13}C ssNMR spectra, and comparing the integrals of the carbohydrate C_1 (integral from 92.0 to 111.0 ppm, see the spectral regions highlighted in purple in **Fig. 1e** and **Supplementary Fig. 16**) and the amino acid carbonyl resonances (integral from 165.0 to 185.0 ppm, grey region in **Supplementary Fig. 16**). We measured a carbohydrate/protein molar ratio of 1:3 (adjusted to 1:2 considering the amount of Asp and Glu with their carboxyl end group). A significant part of the cell wall is thus composed of glycans, in agreement with the literature²⁷ and SDS-PAGE (**Fig. 2a**). Non-destructive qualitative assessment of the glycan/protein ratio is made available using simple 1D ^{13}C ssNMR, while it is challenging for colorimetric or MS approaches known to be destructive, especially for complex eukaryotic samples.

The experiment on **page 11** of the SI (comparison of cryogenic and room temperature (283 K) SQ-DQ INADEQUATE spectra) is now described on **page 24** and a link to the figure in the SI has been added:

We also assigned 339 peaks corresponding to 87 different spin systems in proteins, among 18 different amino acids. All of them were quantified using both HPLC and ssNMR with a method that accounts for overlapping peaks (Poulhazan et al. 2021) (Fig. 3a and Supplementary Fig. 8, and Tables 8-11).

On page 24 of the manuscript, we have also included the discussion on the protein secondary structure:

Chemical shift differences between polyproline II (PPII) helices and random coils are still hard to detect using these tools, especially since PPII helices remain underexplored by NMR. In our sample for instance, Ile, Lys, Thr and Val would be located in β -strand environments, while Asx, Gly, Pro and Ser would either be in random coils or in PPII helices (Fig. 3a). The propensity of an amino acid to be included in a β -sheet has been shown to be more correlated to its neighboring residues than to its chemical nature (Fujiwara et al. 2012) and, indeed, no correlation could be established between the hydrophobicity or charge of the residues and their β -sheet propensity here. In summary, while we can assert the presence of β -strands and exclude that of α -helices in our sample, we cannot rule out the presence of PPII helices previously reported in HRGPs (Homer et al. 1979, Ferris et al. 2001, Voigt et al. 2003).

Minor: please split the figure S3 into 3 SI figures, to provide larger figures to the readers. In the current form, this figure and the assignments are impossible to read.

We thank the Reviewer for this comment, which improves the readability of **Supplementary Figure 3** (now **Supplementary Fig. 5, 6 and 7**). We have made the corresponding changes in the main text figure reference.

A minor change needed: line 389 is the starting of a new paragraph and lacking the incident

Done.

REVIEWERS' COMMENTS

Reviewer #1 (Remarks to the Author):

In response to my queries, the authors toned down the implication of the mannosylated proteins for the *Chlamydomonas* cell wall architecture, with which I agree.

I hope the authors consider the two editorial comments below before publication.

1. The paragraph in LL.360-376 describes the result of the glycoproteomic analysis. The revised paragraph mixes the results and shortfalls and is not straightforward to read.

I suggest that the given paragraph should be focused on what the results clearly show/demonstrate. Then, provide a separate paragraph clearly stating the shortfalls of the results, followed by the sentences added in the revised manuscript below.

Large cell wall proteins with complex glycosylation patterns are notoriously difficult to detect by this approach without using specific molecular separation techniques, enrichment and targeted detection and analysis^{53,54}, but are amenable to ssNMR methods as those used in this study. To further ascertain the high mannosylation of *C. reinhardtii*'s cell wall, our ssNMR approach could be applied on cell-wall extracts from which mannose residues would be removed by mannosidase and other enzymes⁵⁵. Fluorescence imaging using mannose-specific antibodies followed by enzymatic treatment⁵⁶ could also be considered. "

2. Figure 5 provides the depiction of molecular interactions reported in the manuscript. I found that Fig 5d did not add significant insight not presented in 5c.

In addition, the icons used for HRGP and LWGP represent globular proteins, thereby not reflecting the distinct protein structures expected for the HRGP and LWGP pools.

Specifically, W6 (outer fibrous layer) and W2 (inner fibrous layer, mislabeled as W6 in the figure 5d) are fibrous layers, consisting mostly of PP-II helices. On the other hand, the W4 is a globular layer, sandwiched between W2 and W6, where LWGPs may be populated. According to the secondary

structure profiles of cell wall proteins, LWGPs are mostly in the beta-sheet configuration, which is not elaborated in Figure 5d.

So, if the Figure 5d diagram is presented in the publication,

1) correct the W2, W4, W6 labels

2) contrast fibrous and globular layers

3) offer the author's idea on how to make sense of the finding that the beta-sheet peptide configuration dominates in the Chlamydomonas cell wall proteins.

Reviewer #2 (Remarks to the Author):

I recommend publication of this manuscript in Nat. Comm. The authors have clearly improved the manuscript based on the reviewers comments.

Response to the Reviewers

We would like to thank the Reviewers for the thorough revision of our manuscript. Their comments enabled us to improve our manuscript, and more specifically the **Figure 5** and the paragraph on pages 10-11. All comments have been considered within the document. Text modifications were not highlighted in the manuscript following the editorial checklist.

Reviewer #1 (Remarks to the Author)

In response to my queries, the authors toned down the implication of the mannosylated proteins for the *Chlamydomonas* cell wall architecture, with which I agree.

We would like to thank again Reviewer #1 for the comments, which really helped us improve our manuscript throughout the revision process.

I hope the authors consider the two editorial comments below before publication.

1. The paragraph in LL.360-376 describes the result of the glycoproteomic analysis. The revised paragraph mixes the results and shortfalls and is not straightforward to read.

I suggest that the given paragraph should be focused on what the results clearly show/demonstrate. Then, provide a separate paragraph clearly stating the shortfalls of the results, followed by the sentences added in the revised manuscript below.

"Large cell wall proteins with complex glycosylation patterns are notoriously difficult to detect by this approach without using specific molecular separation techniques, enrichment and targeted detection and analysis^{53,54}, but are amenable to ssNMR methods as those used in this study. To further ascertain the high mannosylation of *C. reinhardtii*'s cell wall, our ssNMR approach could be applied on cell-wall extracts from which mannose residues would be removed by mannosidase and other enzymes⁵⁵. Fluorescence imaging using mannose-specific antibodies followed by enzymatic treatment⁵⁶ could also be considered. "

We changed the paragraph on pages 10-11 accordingly to the Reviewer's comment.

The paragraph was:

"The high abundance of Man in C. reinhardtii's cell wall suggests the presence of O-glycosylation to Thr. To verify this hypothesis, a cell-wall protein extract was digested with trypsin and enriched by Concanavalin A lectin weak affinity chromatography (ConA-LWAC)^{51,52}. Our glycoproteomic analysis identified 9 unique glycopeptide sequences from 6 proteins (Supplementary Table 14) using higher-energy collisional activation (HCD) and electron-transfer dissociation (ETciD) MS. Although none of the identified proteins is annotated as a cell wall protein, it should be noted that the approach described above favors abundant, small, soluble proteins with simple glycosylation patterns."

It now reads (lines 291 to 308):

“The high abundance of Man in *C. reinhardtii*'s cell wall suggests the presence of O-glycosylation to Thr. To verify this hypothesis, a cell-wall protein extract was digested with trypsin and enriched by Concanavalin A lectin weak affinity chromatography (ConA-LWAC)^{51,52}. Our glycoproteomic analysis identified 9 unique glycopeptide sequences from 6 proteins (Supplementary Table 14) using higher-energy collisional activation (HCD) and electron-transfer dissociation (ETciD) MS. Although none of the identified proteins is annotated as a cell wall protein, it should be noted that the approach used favors abundant, small, soluble proteins with simple glycosylation patterns. Nevertheless, these glycoproteomic results suggest an O-linked glycosylation of Ser and Thr with hexose modifications, likely α -linked Man or Man₂, thus demonstrating the presence of biosynthetic machineries in *Chlamydomonas* capable of catalyzing these modifications.

Large cell wall proteins with complex glycosylation patterns are notoriously difficult to detect by the approach described above, without using specific molecular separation techniques, enrichment and targeted detection and analysis^{53,54}. They are however amenable to ssNMR methods as those used in this study. To further ascertain the high mannosylation of *C. reinhardtii*'s cell wall, our ssNMR approach could be applied on cell-wall extracts from which mannose residues would be removed by mannosidase and other enzymes⁵⁵. Fluorescence imaging using mannose-specific antibodies followed by enzymatic treatment⁵⁶ could also be considered.”

2. **Figure 5** provides the depiction of molecular interactions reported in the manuscript. I found that **Fig 5d** did not add significant insight not presented in **5c**.

In addition, the icons used for HRGP and LWGP represent globular proteins, thereby not reflecting the distinct protein structures expected for the HRGP and LWGP pools.

Specifically, W6 (outer fibrous layer) and W2 (inner fibrous layer, mislabeled as W6 in the figure 5d) are fibrous layers, consisting mostly of PP-II helices. On the other hand, the W4 is a globular layer, sandwiched between W2 and W6, where LWGPs may be populated. According to the secondary structure profiles of cell wall proteins, LWGPs are mostly in the beta-sheet configuration, which is not elaborated in **Figure 5d**.

So, if the **Figure 5d** diagram is presented in the publication,

- 1) correct the W2, W4, W6 labels
- 2) contrast fibrous and globular layers
- 3) offer the author's idea on how to make sense of the finding that the beta-sheet peptide configuration dominates in the *Chlamydomonas* cell wall protein.

We thank the Reviewer for this comment. We agree that the **Fig. 5d** did not show information significantly different compared to **Fig. 5c**. We changed this scheme to include structural information (given by chemical shift analysis) while tentatively assign LWGP and HRGP to the W4 and W2/W6 layers, as now mentioned in **Figure 5d** caption. We also represented the fibrous shape of the W2/W6 layers containing Polyproline II helical HRGPs while having a larger W4 layer containing LWGPs.